# High-endothelial cell-derived S1P regulates dendritic cell localization and vascular integrity in the lymph node

**Szandor Simmons[1,2,3]\*, Naoko Sasaki[4], Eiji Umemoto[2,4], Yutaka Uchida[1,3], Shigetomo Fukuhara[5], Yusuke Kitazawa[6], Michiyo Okudaira[7], Asuka Inoue[7], Kazuo Tohya[8], Keita Aoi[1,2,3], Junken Aoki[7], Naoki Mochizuki[5], Kenjiro Matsuno[6], Kiyoshi Takeda[2,4], Masayuki Miyasaka[2,9,10], Masaru Ishii[1,2,3]**

[1]Department of Immunology and Cell Biology, Graduate School of Medicine and Frontier Biosciences, Osaka University, Osaka, Japan; [2]WPI Immunology Frontier Research Center, Osaka University, Osaka, Japan; [3]JST CREST, Tokyo, Japan; [4]Department of Microbiology and Immunology, Laboratory of Immune Regulation, Graduate School of Medicine, Osaka University, Osaka, Japan; [5]Department of Cell Biology, National Cerebral and Cardiovascular Center Research Institute, Osaka, Japan; [6]Department of Anatomy (Macro), Dokkyo Medical University, Tochigi, Japan; [7]Laboratory of Molecular and Cellular Biochemistry, Graduate School of Pharmaceutical Sciences, Tohoku University, Miyagi, Japan; [8]Department of Anatomy, Kansai University of Health Sciences, Osaka, Japan; [9]MediCity Research Laboratory, University of Turku, Turku, Finland; [10]Interdisciplinary Program for Biomedical Sciences, Institute for Academic Initiatives, Osaka University, Osaka, Japan

**\*For correspondence:**
simmons@ifrec.osaka-u.ac.jp

**Competing interests:** The authors declare that no competing interests exist.

**Abstract** While the sphingosine-1-phosphate (S1P)/sphingosine-1-phosphate receptor-1 (S1PR1) axis is critically important for lymphocyte egress from lymphoid organs, S1PR1-activation also occurs in vascular endothelial cells (ECs), including those of the high-endothelial venules (HEVs) that mediate lymphocyte immigration into lymph nodes (LNs). To understand the functional significance of the S1P/S1PR1-$G_i$ axis in HEVs, we generated *Lyve1;Spns2*$^{\Delta/\Delta}$ conditional knockout mice for the S1P-transporter Spinster-homologue-2 (SPNS2), as HEVs express LYVE1 during development. In these mice HEVs appeared apoptotic and were severely impaired in function, morphology and size; leading to markedly hypotrophic peripheral LNs. Dendritic cells (DCs) were unable to interact with HEVs, which was also observed in *Cdh5*$^{CRE-ERT2}$;*S1pr1*$^{\Delta/\Delta}$ mice and wildtype mice treated with S1PR1-antagonists. Wildtype HEVs treated with S1PR1-antagonists in vitro and *Lyve1*-deficient HEVs show severely reduced release of the DC-chemoattractant CCL21 in vivo. Together, our results reveal that EC-derived S1P warrants HEV-integrity through autocrine control of S1PR1-$G_i$ signaling, and facilitates concomitant HEV-DC interactions.
DOI: https://doi.org/10.7554/eLife.41239.001

# Introduction

Lymph nodes (LNs) are essential sites for maturation, activation, homeostatic expansion, and tolerance induction of lymphocytes (*Girard et al., 2012*; *von Andrian and Mempel, 2003*), and serve as an immunological interface between the blood and lymph circulatory systems. Their architecture allows them to act as a central filter for the lymph, transporting antigens from the periphery into the LNs. In LNs antigens are recognized by recirculating naïve lymphocytes that immigrate from the

blood through specialized postcapillary venules known as high-endothelial venules (HEVs) (*Girard et al., 2012*; *Miyasaka and Tanaka, 2004*). Naïve lymphocytes scan through paracortical and follicular areas in the LN where they are activated by antigens, subsequently proliferate, and where they develop effector and memory functions. With the acquired competence in pathogen recognition and clearance, effector and memory T- and B-cells patrol through peripheral tissues via peripheral blood and are recruited back into the same or other LNs in the steady state. Lymphocytes remain in circulation until recruited to sites of inflammation, where they facilitate immediate and specific adaptive immune responses locally and thereby provide efficient immune surveillance (*Rosen, 2004*). The entry of naïve T-cells and dendritic cell (DC) precursors from blood, or of effector or memory T-cells and activated DCs from afferent lymphatics into LNs, has been shown to be critically dependent on the interaction of chemokines CCL21 and CCL19 and their receptor CCR7, expressed on lymphocytes or DCs (*Förster et al., 2012*). Importantly, CCL21 is abundantly produced and secreted by high-endothelial cells and accumulates in the perivascular sheath of the HEVs (*Yang et al., 2007*; *Gunn et al., 1998*). Within the HEV lumen, CCL21 facilitates adhesion of rolling lymphocytes and initiates transendothelial migration of T-cells and DC precursors into the LNs (*Yang et al., 2007*; *Stein et al., 2000*). On the other hand, the egress of lymphocytes from the LNs into the lymphatic vasculature is controlled by the lipid-mediator sphingosine-1-phosphate (S1P). Following high concentrations (~µM) of S1P in the circulatory fluids, S1P-receptor-1 (S1PR1) expressing cells leave S1P-reduced (~nM) interstitial sites into the lymphatic vasculature and subsequently migrate to the periphery. Through genetic approaches, targeting the enzymes responsible for S1P production, sphingosine kinase-1 (SPHK1) and sphingosine kinase-2 (SPHK2), both blood endothelial cells (BECs) and lymphatic endothelial cells (LECs) were identified as important sources for providing S1P in blood and lymph, respectively (*Pappu et al., 2007*; *Pham et al., 2010*).

Although lymphatic vessels share some genetic and phenotypic characteristics with the blood vasculature, for example the platelet endothelial cell adhesion molecule (PECAM-1) or the plasmalemma vesicle associated protein 1 (PV-1), their unique gene expression profile reflects differences from lymphatic and blood endothelium. The expression of endoglin or neuropilin-1 (NRP-1) is specific for the blood endothelium while podoplanin (PDPN), lymphatic endothelial hyaluronan receptor-1 (LYVE1), vascular endothelial growth factor (VEGF) receptor-3 (VEGFR-3) and prospero related homeobox 1 (PROX-1) are unique for the lymphatic vasculature in the adult (*Adams and Alitalo, 2007*). In addition to lymphatic vessels and sinuses, LNs have unique blood vessel microarchitecture with venules branching into a characteristic venular tree from the small postcapillary venules in the paracortex, which are called HEVs, to the large collecting venule (*von Andrian, 1996*). The HEVs have a characteristic plump cuboidal morphology and a thick basal lamina with a prominent perivascular sheath. Evidence has been provided that this typical morphology of high–endothelial cells allows the continuous and transient subendothelial accumulation of T- and B-cells in clusters of immigrating lymphocytes from blood to LNs in HEVs (*Mionnet et al., 2011*). High-endothelial cells also have specific molecular fingerprints that characterize the function of HEVs. In particular, they express the L-selectin ligand peripheral node addressin (PNAd) or sialomucins, which represent a family of sulphated, fucosylated and sialylated glycoproteins, including GlyCAM-1, CD34, podocalyxin, endomucin and nepmucin (*Miyasaka and Tanaka, 2004*; *Rosen, 2004*). HEVs are heterogeneous, and the expression of addressins is unique to their localization and developmental stage. For instance, mucosal addressin (MAdCAM-1), a ligand for $\alpha4\beta7$ integrin, can be detected during ontogeny and is preferentially expressed in secondary lymphoid organs of mucosal associated lymphoid tissues. MAdCAM-1 is replaced quickly after birth by perinatally expressed PNAd in peripheral LNs (pLNs) (*Miyasaka and Tanaka, 2004*; *Mebius et al., 1996*). The integrity, phenotype and function of HEVs appear dependent on cellular interactions with neighbouring cells and the availability of angiogenic factors from them. Depletion of DCs in vivo led to the downregulation of PNAd and HEV-specific genes (*Moussion and Girard, 2011*) that is Glycam-1, FucT-VII and Chst4, the latter two encoding for the HEV-unique enzymes fucosyltransferase-7 and N-acetylglucosamine 6-O-sulphotransferase 2 (GlcNAc6ST-2), respectively (*Rosen, 2004*). These drastic phenotypical changes of HEVs resulted in impaired lymphocyte immigration to LNs and were explained by interrupted stimulation of lymphotoxin-$\beta$ receptor (LT$\beta$r) signalling that is evoked by LT$\alpha$1$\beta$2, provided by DCs (*Moussion and Girard, 2011*; *Browning et al., 2005*). In addition, DCs closely associate with fibroblastic reticular cells (FRCs) and present LT$\alpha$1$\beta$2 to LT$\beta$r expressed on FRCs which, in turn, leads to the production of VEGF as an angiogenic factor to HEVs (*Wendland et al., 2011*; *Chyou et al., 2011*; *Kumar et al.,*

*2015*). FRCs and platelets also regulate HEV integrity. Mice lacking FRC-podoplanin or platelet C-type lectin-like receptor-2 (CLEC2) have abnormal HEVs and show spontaneous bleeding into LNs (*Herzog et al., 2013*). Moreover, podoplanin has been identified as an activating ligand for CLEC2 that, upon interaction, leads to S1P-secretion from platelets (*Herzog et al., 2013*).

We reported recently that BECs secrete S1P via the transporter Spinster-homolog-2 (SPNS2) (*Fukuhara et al., 2012*), which had been identified by us and others as a specific S1P transporter in zebra fish (*Kawahara et al., 2009*; *Osborne et al., 2008*). The S1P secretion from BECs by SPNS2 contributes to about 50% of the total S1P in the blood plasma whereas hematopoietic cells, for example platelets and erythrocytes, release S1P in a SPNS2-independent manner (*Fukuhara et al., 2012*; *Hisano et al., 2012*). Therefore, influenced by the activities and distribution of degradative and biosynthetic enzymes, ECs account for the formation of an S1P gradient responsible for lymphocyte egress from thymus and secondary lymphoid organs (*Fukuhara et al., 2012*; *Hisano et al., 2012*; *Mendoza et al., 2012*; *Nagahashi et al., 2013*; *Nijnik et al., 2012*). Here we found that *Spns2* is expressed in high-endothelial cells that have been described as showing constitutively active S1PR1-$G_i$ signalling (*Kono et al., 2014*). Hence, we hypothesized that high-endothelial cells also secrete S1P that, in turn, acts on HEVs to regulate their function, particularly to regulate lymphocyte migration across HEVs. Specific and conditional gene targeting in high-endothelial cells has been reported in a transgenic mouse line expressing Cre recombinase under the transcriptional control of the gene encoding HEV-expressed GlcNAc6ST-2 (29). In this study, we find LYVE1 expressed in high-endothelial cells in foetal stages, while HEVs lack the expression of LYVE1 in juvenile and adult mice. Therefore, we generated *Spns2*-deficient *Lyve1;Spns2*$^{\Delta/\Delta}$ mice by deleting loxP-flanked *Spns2* in *Spns2*$^{f/f}$ mice through intercrossing with *Lyve1*$^{CRE}$ mice (*Pham et al., 2010*; *Fukuhara et al., 2012*). In accordance with our hypothesis we found impaired HEVs and strongly reduced lymphocyte immigration leading to the development of hypotrophic pLNs in *Lyve1;Spns2*$^{\Delta/\Delta}$ mice. In this study, we provide evidence that S1P secreted by SPNS2 from high-endothelial cells triggers autocrine activation of S1PR1-$G_i$-signalling in HEVs. Furthermore, S1PR1-$G_i$-signalling regulates high-endothelial cell survival, HEV-integrity and coincidentally CCL21 production and release from high-endothelial cells. Consequently, this negatively influences HEV-DC interactions necessary for normal morphology and function of HEVs, which allows controlled lymphocyte immigration into pLNs.

## Results

### Lyve1$^{CRE}$-mediated Spns2-deletion in endothelial cells

The spleen and thymus of *Lyve1;Spns2*$^{\Delta/\Delta}$ mice were unaltered in their tissue architecture and size, whereas in comparison to wildtype *Spns2*$^{f/f}$ mice, pLNs are hypotrophic (*Figure 1 (A)*). *Lyve1*-specific deletion of *Spns2* did not affect S1P levels in blood, but did reduce the S1P concentration in lymph fluid to only 14.7% of that seen in lymph of *Spns2*$^{f/f}$ mice (*Figure 1 (B)*). Nevertheless, sphingosine and glycerol-based lysophospholipid levels in both blood and lymph were comparable between *Spns2*-deficient and control mice (*Figure 1 (B)* and *Figure 1—figure supplement 1*). Previously we have shown that *Spns2*-deficiency in global *Spns2*$^{-/-}$ mice and conditional *Tie2-Spns2*$^{\Delta/\Delta}$ mice results in a significant reduction of S1P-levels in the blood (*Fukuhara et al., 2012*). In consequence, impairment of the chemotactic S1P-gradient between thymic interstitium and blood vasculature resulted in a reduced egress of S1PR1-expressing T-cells and an accumulation of CD4$^+$ and CD8$^+$ single-positive (SP) T-cells in the thymus (*Fukuhara et al., 2012*). Nonetheless, frequencies and total cell numbers of CD4$^+$ and CD8$^+$ SP T-cells in the thymus of *Lyve1;Spns2*$^{\Delta/\Delta}$ mice were comparable to those of *Spns2*$^{f/f}$ mice (*Figure 1—figure supplement 2 (A)*), whereas the hypotrophic pLNs of *Lyve1;Spns2*$^{\Delta/\Delta}$ mice showed a strong reduction in the total number of cells to only 20.1% of CD4$^+$ and 21.7% of CD8$^+$ SP T-cells, and 59% of mature recirculating (rec.) B-cells (CD19$^+$/CD23$^+$/IgD$^+$; *Figure 1 (C–D)*). These results indicate that *Lyve1*$^{CRE}$ mediated *Spns2* deletion may prevent entry of recirculating lymphocytes to pLNs. Recirculating B- and T-cell populations were strongly decreased throughout various lymphoid organs in *Lyve1;Spns2*$^{\Delta/\Delta}$ mice (*Figure 1—figure supplement 2 (A-E)*). In the spleen of *Lyve1;Spns2*$^{\Delta/\Delta}$ mice, follicular (FO) B-cells were reduced to 43.5%, whereas marginal zone (MZ) B-cells were increased to 55% (*Figure 1—figure supplement 2 (D-E)*). Furthermore, in the thymus, BM and spleen of *Lyve1;Spns2*$^{\Delta/\Delta}$ mice we detected elevated frequencies of apoptotic

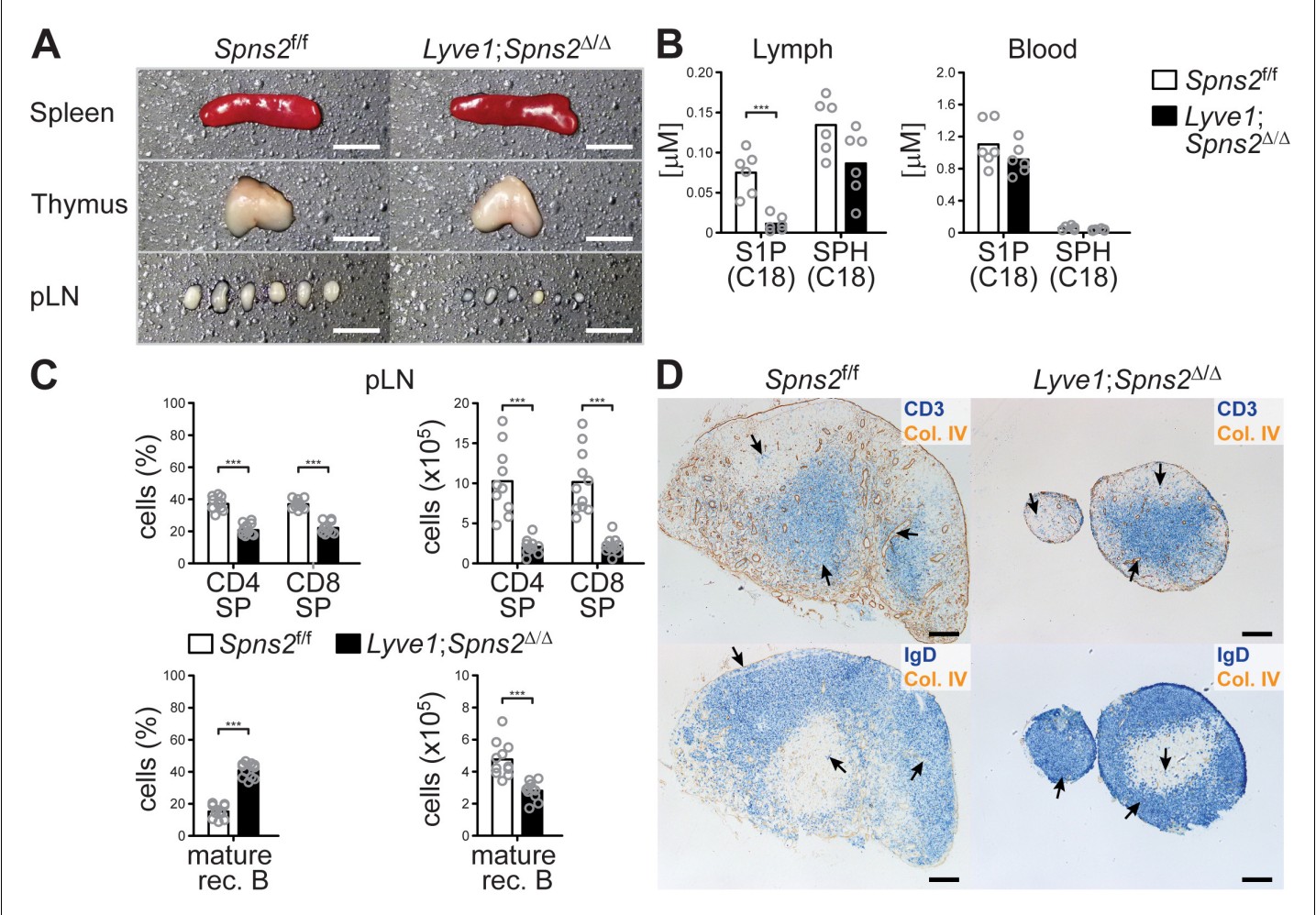

**Figure 1.** *Lyve1*[CRE]-mediated *Spns2*-deletion in endothelial cells causes hypotrophy in pLNs. (**A**) Macroscopic view of spleen, thymus and pLNs of wildtype *Spns2*[f/f] (left) and *Lyve1;Spns2*[Δ/Δ] (right) mice. (**B**) Quantification of S1P and sphingosine (SPH) concentrations in lymph and blood. (**C**) FACS analysis of CD4[+] and CD8[+] SP T-cells (top) and mature rec. B-cells (bottom) of pLNs of *Spns2*[f/f] and *Lyve1;Spns2*[Δ/Δ] mice. (**D**) Light microscopy of pLNs of *Spns2*[f/f] (left) and *Lyve1;Spns2*[Δ/Δ] (right) mice for CD3[+] T-cells (top, blue) and IgD[+] mature rec. B-cells (bottom, blue) counterstained for collagen-IV[+] (brown) tissue frameworks. Each circle (**B, C**) represents an individual mouse; bars indicate the mean. Scale bars, 0.2 cm (**A**) or 200 µm (**D**). ***$p<0.0005$ (two-tailed unpaired Student's *t*-test (**B, C**)). Data are representative for six mice per group (**A, B**), for 2x inguinal, 2x brachial and 2x axial LNs of six mice per group (**D**) or are pooled from three independent experiments (**C**) with n = 3 or n = 4 mice per group.
DOI: https://doi.org/10.7554/eLife.41239.002

The following figure supplements are available for figure 1:

**Figure supplement 1.** *Lyve1*[CRE] mediated *Spns2*-deficiency does not affect glycerol-based lysophospholipid levels, representatively shown for C18:1 species (other species (not shown) were also checked and found to be not different), in lymph and blood.
DOI: https://doi.org/10.7554/eLife.41239.003

**Figure supplement 2.** Recirculating lymphocyte populations are impaired throughout various lymphatic tissues in *Lyve1;Spns2*[Δ/Δ] mice.
DOI: https://doi.org/10.7554/eLife.41239.004

lymphocytes, whereas the frequencies of apoptotic B- and T-cells were reduced in pLNs of *Lyve1; Spns2*[Δ/Δ] mice (***Figure 1—figure supplement 2 (F)***). These results are compatible with the hypothesis that LYVE1[+] LECs make a significant contribution to lymph S1P-levels by secreting S1P via SPNS2 into the lymphatics and thereby control lymphocyte egress from secondary lymphoid organs into the lymphatic system.

## Reduced lymphocyte immigration into pLNs and impaired lymphocyte egress into the lymphatic system in Lyve1;Spns2$^{\Delta/\Delta}$ mice

Consistent with the strong reduction of lymph S1P-levels (*Figure 1 (B)*), flow cytometric analyses of the lymph collected from the *cisterna chyli* of *Lyve1;Spns2*$^{\Delta/\Delta}$ mice confirmed a complete absence of recirculating lymphocytes (*Figure 2 (A)*), indicating severe impairment of lymphocyte-egress into the lymphatics from pLNs. Moreover, the drastic difference in organ volume (*Figure 1 (A)*) and lymphocyte homeostasis (*Figure 1 (B–D)*) between pLNs of *Lyve1;Spns2*$^{\Delta/\Delta}$ mice suggests that lymphocyte immigration from the blood into pLNs was affected, even if the numbers of lymphocytes in circulation are influenced by their reduced viability (*Figure 1—figure supplement 2 (B-F)*). Therefore, we tested short-term lymphocyte trafficking to lymphoid tissues by adoptively transferring wildtype congenic CD45.1$^{+}$ splenocytes *i.v.* into *Spns2*$^{f/f}$ and *Lyve1;Spns2*$^{\Delta/\Delta}$ mice (*Figure 2 (B)*). Flow cytometric analyses recorded two hours after injection of cells showed that immigration into pLNs of *Lyve1; Spns2*$^{\Delta/\Delta}$ mice was ~5.2 fold less efficient than into pLNs of *Spns2*$^{f/f}$ mice (*Figure 2 (C)*). Given that cellular immigration into pLNs was strongly reduced, we next sought to determine if *Lyve1* specific ablation of *Spns2* had an effect on HEVs and found that development of PNAd$^{+}$ HEVs appear to be severely compromised in pLNs of *Lyve1;Spns2*$^{\Delta/\Delta}$ mice (*Figure 2 (D)*). In agreement with this observation in IHC, the frequencies of CD45$^{-}$/CD31$^{+}$/PNAd$^{+}$ high-endothelial cells isolated from pLNs of *Lyve1;Spns2*$^{\Delta/\Delta}$ were ~2.3 fold reduced and total cell numbers were ~10.3 fold decreased in comparison to the controls when analysed by FACS (*Figure 2 (E)*). In order to functionally address the efficiency of the egress of lymphocytes from pLNs, congenic eGFP$^{+}$ splenocytes were adoptively transferred into *Spns2*$^{f/f}$ and *Lyve1;Spns2*$^{\Delta/\Delta}$ recipient mice. After an equilibration period of 48 hr surface integrins on circulating lymphocytes were saturated with anti-$\alpha_L$ and anti-$\alpha_4$ antibodies, as previously described (*Lo et al., 2005*). This results in reduced lymphocyte arrest under physiological shear at the endothelial cell wall. Therefore, lymphocyte immigration across HEVs was blocked at t = 0 hr and lymphocyte egress rates could be quantified 20 hr later (t = 20 hr, *Figure 2 (F)*). The numbers of adoptively transferred eGFP$^{+}$ cells that were present 20 hr after integrin blockade in pLNs of *Spns2*$^{f/f}$ mice were strongly reduced in comparison to the cell numbers detected at 0 hr (*Figure 2 (G)*). However, the cell numbers in pLNs of *Lyve1;Spns2*$^{\Delta/\Delta}$ mice at 20 hr where unaltered when compared to those at 0 hr (*Figure 2 (G)*). These results functionally show that lymphocyte egress is reduced in pLNs of *Lyve1;Spns2*$^{\Delta/\Delta}$ mice in comparison to that measured in pLNs of *Spns2*$^{f/f}$ mice (*Figure 2 (G)*). In summary, our data demonstrate, unexpectedly, that *Spns2*-deficiency in LYVE1$^{+}$endothelial cells led to severe impairment in function, morphology and size of PNAd$^{+}$ HEVs, resulting in reduced immigration of lymphocytes which, in turn, led to the development of hypotrophic pLNs of *Lyve1;Spns2*$^{\Delta/\Delta}$ mice. Moreover, lymphocyte egress from pLNs into the lymphatic system is impaired in *Lyve1;Spns2*$^{\Delta/\Delta}$ because of the impaired S1P secretion via SPNS2 by LYVE1$^{+}$ LECs.

## Spns2-deficiency in HEVs of Lyve1;Spns2$^{\Delta/\Delta}$ mice

The abnormal morphology and function of HEVs and the concomitant difference in lymphocyte immigration to pLNs of *Lyve1;Spns2*$^{\Delta/\Delta}$ mice prompted us to investigate *Lyve1*$^{CRE}$ mediated gene deletion in HEVs. To this end, we intercrossed *Lyve1*$^{CRE}$-mice to mice carrying *tdTomato* preceded by a *LoxP*-flanked transcriptional stop in the *Rosa26* locus (*Madisen et al., 2010*). We revealed that more than 90% of HEVs of pLNs simultaneously expressed PNAd and tdTOMATO in *Lyve1;tdTomato* mice, but did not express LYVE1 at detectable levels on the cell surface (*Figure 2—figure supplement 1 (A-B)*). Furthermore, a comparable frequency of LECs isolated from *Lyve1;tdTomato* mice expressed the tdTOMATO reporter protein (*Figure 2—figure supplement 1 (A-B)*). Flow cytometric analyses of PNAd$^{+}$ high-endothelial cells isolated from pLNs of adult *Spns2*$^{f/f}$ and *Lyve1;Spns2*$^{\Delta/\Delta}$ mice also confirmed the absence of *Lyve1* expression on the surface of HEVs (*Figure 2—figure supplement 1 (C)*). However, quantitative RT-PCR analyses confirmed the deletion of *Spns2* in purified CD45$^{-}$/CD31$^{+}$/PNAd$^{+}$ high-endothelial cells and CD45$^{-}$/LYVE1$^{+}$ LECs in pLNs of *Lyve1;Spns2*$^{\Delta/\Delta}$ mice (*Figure 2—figure supplement 1 (D)*). We therefore hypothesized that LYVE1 is expressed in high-endothelial progenitor cells during ontogeny, and that LYVE1 expression downregulates with maturation of HEVs after birth. Hence, we next examined whether LYVE1-expression can be found on high-endothelial cell progenitors and PNAd$^{+}$ HEVs in inguinal LNs (iLNs) of WT embryos of E16.5 and E18.5. iLNs of E16.5 expressed LYVE1 weakly on MAdCAM-1$^{+}$/PNAd$^{-}$endothelial cells that are

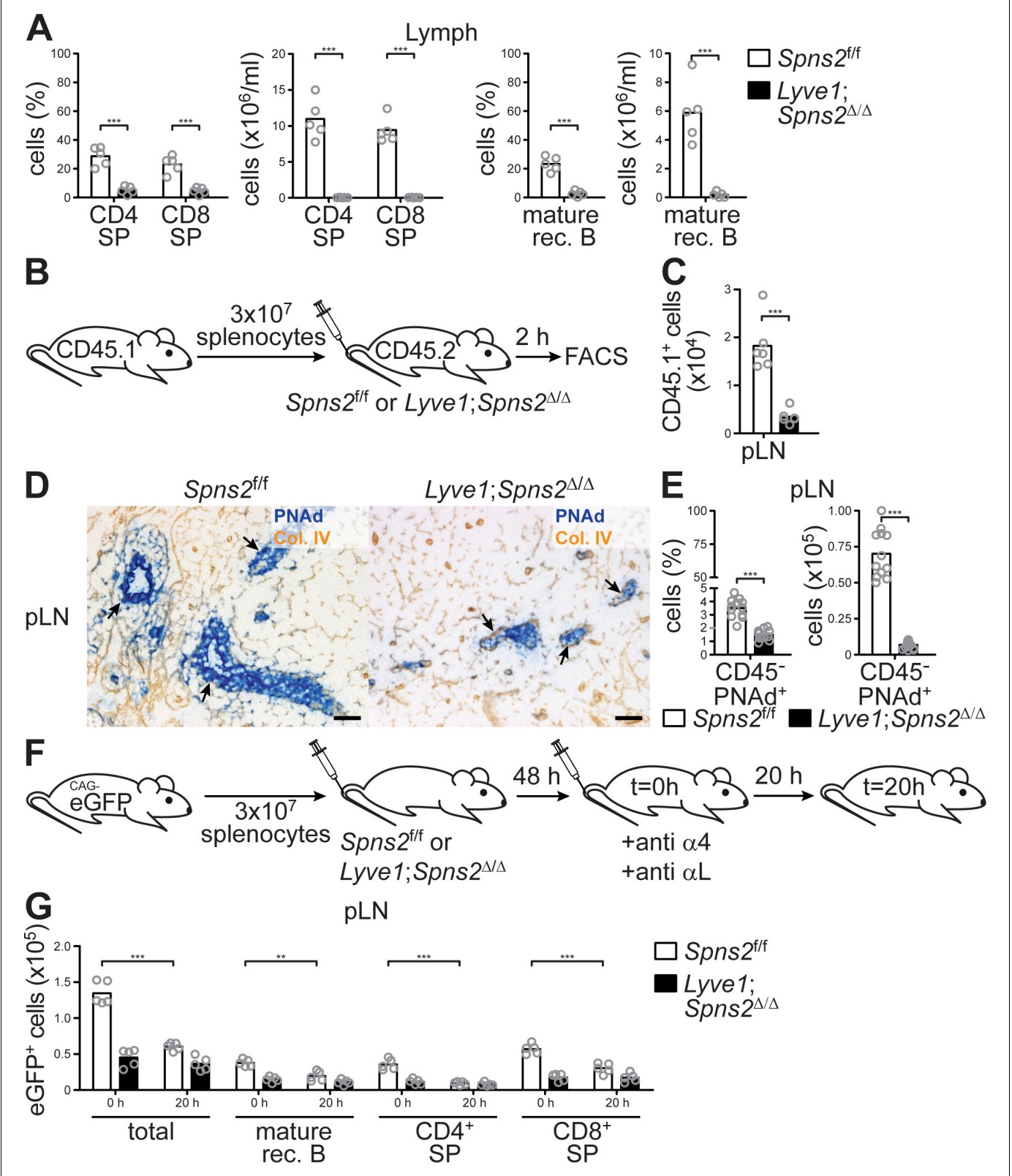

**Figure 2.** The immigration of lymphocytes into pLNs and their egress into the lymphatic system is severely impaired in Lyve1;$Spns2^{\Delta/\Delta}$ mice. (**A**) FACS analysis of CD4$^+$ and CD8$^+$ SP T-cells (left) and mature rec. B-cells (right) of lymph fluid isolated from the *cisterna chyli* of $Spns2^{f/f}$ and $Lyve1;Spns2^{\Delta/\Delta}$ mice. (**B**) Experimental flow-chart of short-terming homing assays to quantify lymphocyte immigration into pLNs. (**C**) FACS analysis of total congenic

*Figure 2 continued on next page*

Figure 2 continued

CD45.1$^+$ cells in pLNs two hours upon injection of WT splenocytes into *Spns2*$^{f/f}$ and *Lyve1;Spns2*$^{Δ/Δ}$ mice. (D) Light microscopy of frozen sections of pLNs of *Spns2*$^{f/f}$ (left) and *Lyve1;Spns2*$^{Δ/Δ}$ (right) mice for PNAd$^+$ HEVs (blue) counterstained for collagen-IV$^+$ (brown). (E) FACS analysis of total CD45$^-$/CD31$^+$/PNAd$^+$ high-endothelial cells isolated from pLNs of *Spns2*$^{f/f}$ and *Lyve1;Spns2*$^{Δ/Δ}$ mice. (F) Experimental flow-chart of homing assays to quantify lymphocyte egress from pLNs. (G) Total numbers of congenic eGFP$^+$ cells in pLNs at 0 hr and 20 hr upon injection of anti-α$_4$ / anti-α$_L$ antibodies into *Spns2*$^{f/f}$ and *Lyve1;Spns2*$^{Δ/Δ}$ mice. Each circle (A, C, E, G) represents an individual mouse; bars indicate the mean. Scale bars, 50 μm (D). **p<0.005; ***p<0.0005 (two-tailed unpaired Student's *t*-test (A, C, E, G)). Data are representative for five mice per group pooled from two independent experiments (A) with n = 2 or n = 3 mice per group (A), for six mice per group (D) or are pooled from two (C, G) or three (E) independent experiments with n = 2, n = 3 or n = 4 mice per group.

DOI: https://doi.org/10.7554/eLife.41239.005

The following figure supplement is available for figure 2:

**Figure supplement 1.** *Spns2* is effectively deleted in LECs and HEVs of *Lyve1;Spns2*$^{Δ/Δ}$ mice.

DOI: https://doi.org/10.7554/eLife.41239.006

evocative in localization and morphology to high-endothelial cell progenitors (*Figure 2—figure supplement 1 (E)*, top). Furthermore, triple-positive MAdCAM-1$^+$/PNAd$^+$/LYVE1$^+$ HEVs were found in iLNs of E18.5 (*Figure 2—figure supplement 1 (E)*, bottom). These data strongly support the hypothesis that the progenitors of high-endothelial cells express LYVE1 during ontogeny which results in efficient deletion of *Spns2* in PNAd$^+$ HEVs of pLNs of *Lyve1;Spns2*$^{Δ/Δ}$ mice.

## Dependency of the integrity of PNAd$^+$ HEVs in pLNs on lymph-derived DCs

Given that DCs play a critical role in the maintenance of HEV architecture and function in pLNs (*Moussion and Girard, 2011*; *Wendland et al., 2011*), we analysed DC homeostasis in *Lyve1;Spns2*$^{Δ/Δ}$ mice. Although a ~ 2.8 fold increase of the total CCR7$^-$ conventional migratory DCs (mDCs) per ml blood of *Lyve1;Spns2*$^{Δ/Δ}$ mice possibly reflects an impaired immigration of DCs to pLNs, DC frequencies in pLNs, spleen and BM, did not differ significantly between *Spns2*$^{f/f}$ and *Lyve1;Spns2*$^{Δ/Δ}$ mice (*Figure 3 (A)*, *Figure 3—figure supplement 1 (A)*). However, in comparison to *Spns2*$^{f/f}$ mice total numbers of resident DCs (rDCs) and mDCs in pLNs of *Lyve1;Spns2*$^{Δ/Δ}$ mice were reduced to 65.2% (rDCs) and 58.4% (mDCs) indicating reduced immigration of DCs (*Figure 3 (A)*). Remarkably, surface expression of CCR7 was unaltered in both rDCs and mDCs in *Spns2*$^{f/f}$ and *Lyve1;Spns2*$^{Δ/Δ}$ mice (*Figure 3 (A)*). We observed that endogenous DCs were in close proximity to PNAd$^+$ HEVs in pLNs of *Spns2*$^{f/f}$ mice (*Figure 3 (B)*, *Figure 3—figure supplement 1 (B)*). In contrast, endogenous DCs were absent in areas adjacent to PNAd$^+$ HEVs in pLNs of *Lyve1;Spns2*$^{Δ/Δ}$ mice (*Figure 3 (B)*), thus unlikely to provide angiogenic factors to high-endothelial cells. Transmission electron micrographs (TEM) confirmed the drastic morphological changes in atrophic HEVs in pLNs of *Lyve1;Spns2*$^{Δ/Δ}$ mice (*Figure 3 (C)*). The number of high-endothelial cells of a HEV appeared to be reduced and the height of EC is relatively flat in pLNs of *Lyve1;Spns2*$^{Δ/Δ}$ mice when compared to those in pLNs of *Spns2*$^{f/f}$ mice (*Figure 3 (C)*). Moreover, high-endothelial cells lost their characteristic cuboidal morphology and nuclei of high-endothelial cells appear deformed in comparison to HEVs in pLNs of *Spns2*$^{f/f}$ mice (*Figure 3 (C)*). This prompted us to asses if high-endothelial cells from pLNs of *Lyve1;Spns2*$^{Δ/Δ}$ mice are apoptotic. Indeed, a flow cytometric terminal deoxynucleotidyl ransferase (TdT) dUTP Nick-End Labeling (TUNEL) assay showed a dramatic ~25 fold increase of apoptotic CD45$^-$/CD31$^+$/PNAd$^+$ high-endothelial cells in pLNs isolated from *Lyve1;Spns2*$^{Δ/Δ}$ mice in comparison to the controls (*Figure 3 (D)*). However, high-endothelial cell related PNAd scaffold proteins (GlyCAM-1, CD34, MadCAM-1), glycan and LPA synthetic enzymes (GlcNAc6ST-2, ENPP2) and vasculature associated (CD31, VCAM-1, ICAM-1, VE-cadherin) gene expression were unaltered between *Lyve1;Spns2*$^{Δ/Δ}$ and *Spns2*$^{f/f}$ mice (*Figure 3—figure supplement 1 (C)*). Only a mild reduction in mRNA expression of alpha-(*Girard et al., 2012*; *Miyasaka and Tanaka, 2004*)-fucosyltransferase-VII (FucT-VII) of ~2.75 fold and of lymphotoxin beta receptor (LTBR) of ~1.8 fold could be observed (*Figure 3—figure supplement 1 (C)*). Given the reduced HEV-DC interactions in pLNs of *Lyve1;Spns2*$^{Δ/Δ}$ mice and the impaired integrity of HEVs, we next asked whether *Lyve1*-specific ablation of *Spns2* affected immigration of activated DCs by afferent lymphatics and the control of DC localization around HEVs. For this purpose, we injected fluorescently labelled mature bone-marrow derived DCs (BMDCs) into the footpad of *Spns2*$^{f/f}$ and *Lyve1;Spns2*$^{Δ/Δ}$ mice and investigated their migration into

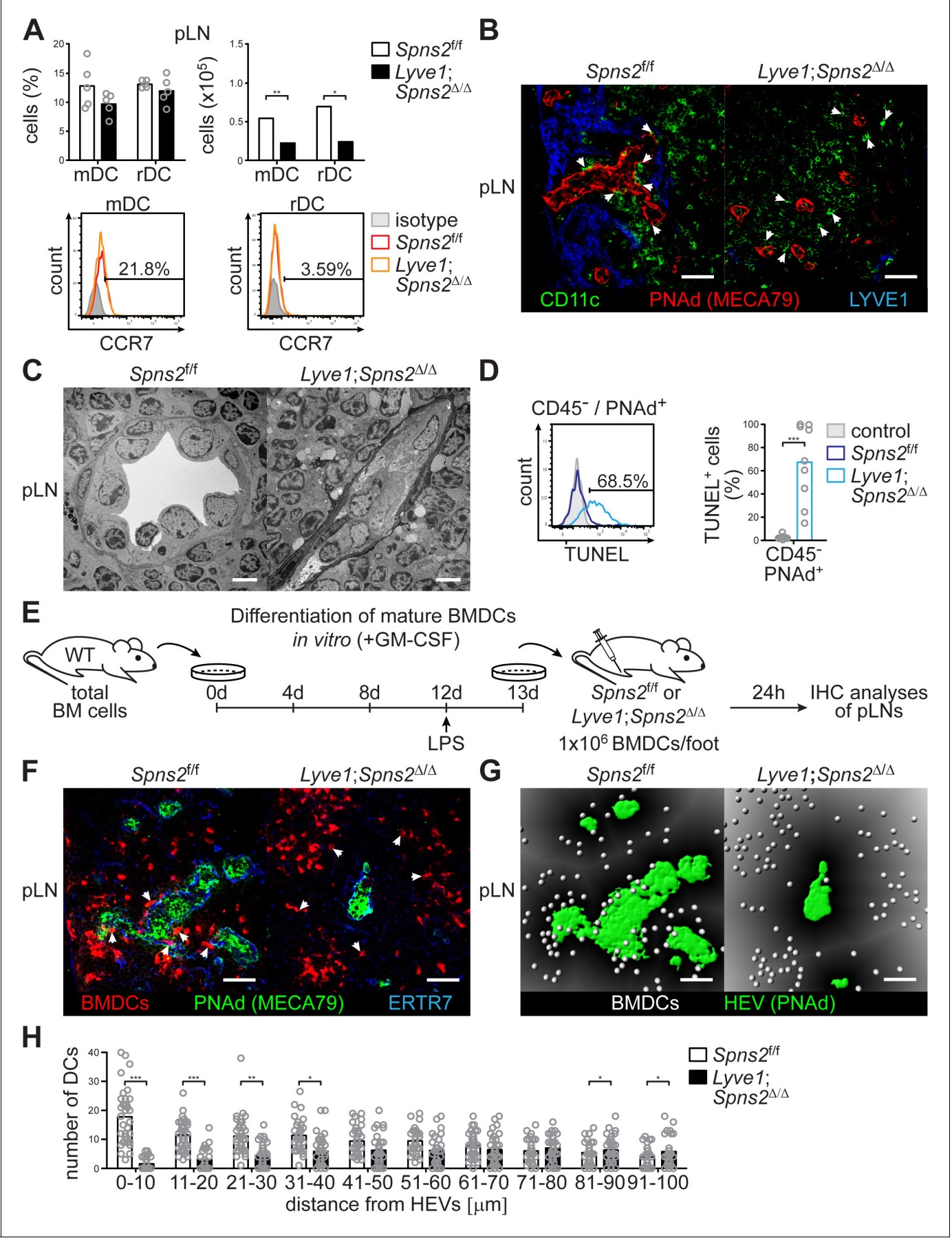

**Figure 3.** SPNS2-derived S1P controls interactions of PNAd$^+$ HEVs with lymph-derived dendritic cells in pLNs. (A) FACS analysis of CCR7-expression on endogenous conventional mDCs (CD3$^-$/CD19$^-$/CD11c$^{int}$/MHC-II$^{hi}$) and rDCs (CD3$^-$/CD19$^-$/CD11c$^{hi}$/MHC-II$^{int}$) isolated from pLNs of $Spns2^{f/f}$ and $Lyve1;Spns2^{\Delta/\Delta}$ mice. (B) Confocal microscopy of pLNs of $Spns2^{f/f}$ (left) and $Lyve1;Spns2^{\Delta/\Delta}$ (right) mice for CD11c$^+$ (green) DCs, PNAd$^+$ (red) HEVs and Lyve-1$^+$ (blue) LECs. (C) TEM images of HEVs in pLNs of $Spns2^{f/f}$ and $Lyve1;Spns2^{\Delta/\Delta}$ mice. (D) Flow-cytometric TUNEL assay on CD45$^-$/CD31$^+$/PNAd$^+$ high-endothelial cells isolated from pLNs of $Spns2^{f/f}$ and $Lyve1;Spns2^{\Delta/\Delta}$ mice. (E) Experimental flow-chart of BMDC-differentiation in vitro, and lymphatic homing assays of footpad injected BMDCs to quantify DC-immigration from afferent lymphatics into pLNs of $Spns2^{f/f}$ and $Lyve1;Spns2^{\Delta/\Delta}$ mice. (F) Confocal microscopy of pLNs of $Spns2^{f/f}$ (left) and $Lyve1;Spns2^{\Delta/\Delta}$ (right) mice for CMTMR$^+$ BMDCs (red), PNAd$^+$ (green) HEVs and ERTR7$^+$ (blue) fibroblastic tissue networks. (G) Visualisation of the automated detection of individual CMTMR$^+$ BMDCs (white spheres) from PNAd$^+$ HEVs (green surface) in pLNs of $Spns2^{f/f}$ (left) and $Lyve1;Spns2^{\Delta/\Delta}$ (right) mice. Grey gradients visualise the distance transformation from HEVs (green surface) defined by PNAd-staining. (H) Total numbers of BMDCs (white spheres in (F)) in distances from 0 µm - 100 µm from HEVs counted in 10 µm radial areas around HEVs in pLNs of $Spns2^{f/f}$ and $Lyve1;Spns2^{\Delta/\Delta}$ mice. Each circle represents an individual mouse (A, D) or total numbers of BMDCs around HEVs in the visual field of a micrograph (H); bars indicate the mean. Scale bars, 5 µm (C), 50 µm (B, F, G). *p<0.05; **p<0.005; ***p<0.0005 (two-tailed unpaired Student's t-test (A, D, H)). Data are representative for six mice per group pooled from two (A, B) or three (D) independent experiments with n = 3 (A) or n = 4 per (D) mice group, for 2x pLNs and 2x iLNs of three mice per group (C), for 36x representative individual sections of 2x analyzed popliteal LNs per mouse pooled from two independent experiments (H) with n = 6 mice per group (H).

DOI: https://doi.org/10.7554/eLife.41239.007

The following figure supplement is available for figure 3:

**Figure supplement 1.** Endogenous DCs do not co-localize with HEVs in pLNs of $Lyve1;Spns2^{\Delta/\Delta}$ mice.

DOI: https://doi.org/10.7554/eLife.41239.008

pLNs 24 hr later (*Figure 3 (E)*). BMDCs immigrated through afferent lymphatics into pLNs of $Lyve1;Spns2^{\Delta/\Delta}$ mice in frequencies comparable to those of the controls. Strikingly, total numbers of homed BMDCs were strongly reduced particularly in a restricted area 40 µm from the basal lamina of HEVs in pLNs of $Lyve1;Spns2^{\Delta/\Delta}$ mice when compared to BMDCs in pLNs of $Spns2^{f/f}$ mice (*Figure 3 (F–H)*). These data are compatible with the idea that the absence of SPNS2-dependent release of S1P from HEVs in pLNs of $Lyve1;Spns2^{\Delta/\Delta}$ mice caused impaired interactions of HEV with activated lymph-derived DCs, making DCs unable to support normal development and function of HEVs via humoral factors such as LTα1β2, which collectively resulted in heavily restricted lymphocyte immigration to pLNs. This hypothesis was supported by the partial rescue of total high-endothelial cell numbers and HEV morphology in pLNs of $Lyve1;Spns2^{\Delta/\Delta}$ mice observed upon 10 weeks of treatment with agonistic anti-LTβr antibody and recombinant LTα1/β2 protein (*Figure 4 (A–D)*). While this treatment was unable to rescue the size of pLNs (*Figure 4 (A–C)*), it significantly increased the total numbers of HEVs and the average PNAd$^+$ area per LN-section (*Figure 4 (B–D)*).

## Impaired HEV-DC interactions induced by S1PR1 antagonists

In order to understand how HEV-DC interactions and the migratory ability of DCs in pLNs are controlled by HEV-derived S1P, we blocked S1PR signalling by the nonspecific S1PR antagonist FTY720 before we injected mature BMDCs into the footpad of wildtype C57BL/6 mice (*Figure 5 (A)*). FTY720 targets four of the five S1PRs (S1PR1 and S1PR3-5) (*Brinkmann et al., 2002*) and should therefore induce impaired HEV-DC interactions of lymph-derived BMDCs in pLNs of recipient mice. Again, 24 hr after footpad injection we observed a severe reduction of homed BMDCs in a restricted area around HEVs within an extended area 70 µm from the basal lamina of HEVs (*Figure 5 (B–D)*), in agreement with the hypothesis that HEV-derived S1P controls DC localization in pLNs. Indeed, HEVs have been described to express S1PR1 and activate strong S1PR1-G$_i$ signalling (*Kono et al., 2014*; *Lee et al., 2014*). In line with these observations our description of impaired HEV-DC interactions upon FTY720 treatment shows that HEVs are highly S1P-responsive. Certainly, mature BMDCs were also found to migrate to high S1P concentration, a phenomenon that correlated to the up-regulation of S1PR1 and S1PR3 (*Czeloth et al., 2005*; *Maeda et al., 2007*). In order to further evaluate the stimulation of specific S1PRs and their relationship in facilitating HEV-DC interactions we took advantage of the S1PR1-specific antagonist W146 and the S1PR3-specific antagonist TY52156. Osmotic pump implantation *i.p.* into wildtype C57BL/6 mice 48 hr prior to footpad injection of mature BMDCs provided constant antagonist levels in recipient mice (*Figure 5 (D)*). Interestingly, abrogation of S1PR1-G$_i$ signalling with W146 also induced impaired HEV-DC interactions in a restricted area within 40 µm from the basal lamina of HEVs in pLNs of recipient mice (*Figure 5 (F)* and *Figure 5—figure supplement 1 (A-B)*). However, application of TY52156, and the concomitant block of S1PR3-signalling, did not affect localization of DCs around HEVs (*Figure 5 (G)* and *Figure 5—figure*

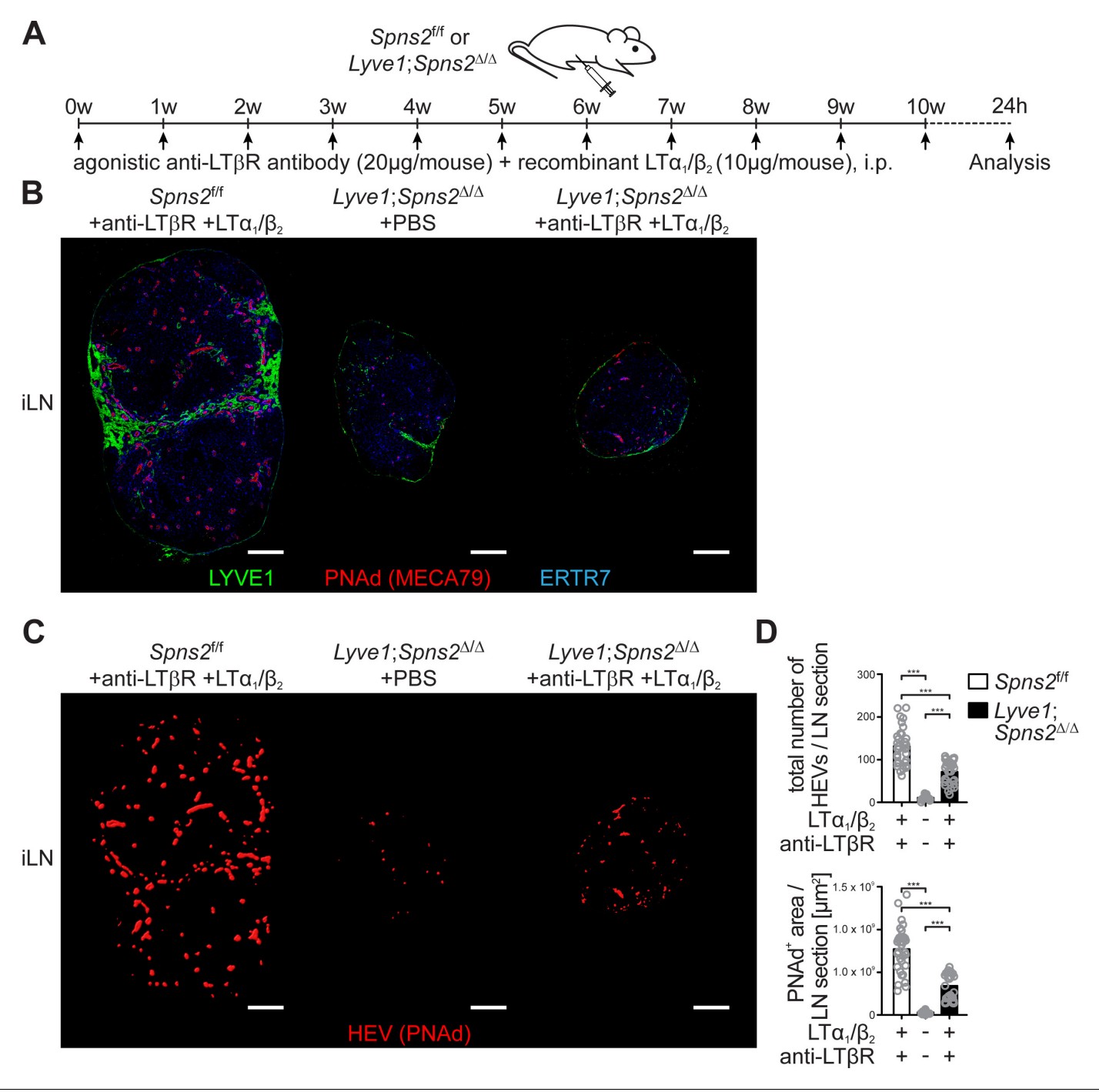

**Figure 4.** Combined anti-LTβr antibody and recombinant LTα1/β2 protein treatment partially rescues total high-endothelial cell numbers and HEV morphology. (A) Experimental flow-chart of PBS or agonistic anti-LTβr antibody (20 µg/mouse) and recombinant LTα1/β2 protein (10 µg/mouse) i.p. injections into $Spns2^{f/f}$ and $Lyve1;Spns2^{\Delta/\Delta}$ mice. (B) Confocal microscopy of iLNs of $Spns2^{f/f}$ mice +anti-LTβr antibody +LTα1/β2 (left), $Lyve1;Spns2^{\Delta/\Delta}$ mice +PBS (mid) and $Lyve1;Spns2^{\Delta/\Delta}$ mice +anti-LTβr antibody +LTα1/β2 (right) mice for LYVE1+ (green) LECs, PNAd+ (red) HEVs and ERTR7+ (blue) fibroblastic tissue networks. (C) Visualisation of the automated detection of PNAd+ HEVs (red surfaces) used for the quantification of the total numbers of HEVs/LN section and the total PNAd+ area/LN section of iLNs of $Spns2^{f/f}$ mice +anti-LTβr antibody +LTα1/β2 (left), $Lyve1;Spns2^{\Delta/\Delta}$ mice +PBS (mid) and $Lyve1;Spns2^{\Delta/\Delta}$ mice +anti-LTβr antibody +LTα1/β2 (right) mice. (D) The total numbers of HEVs/LN section and the total PNAd+ area/LN section extracted from the analyses shown in (B) and (C) of iLNs of $Spns2^{f/f}$ and $Lyve1;Spns2^{\Delta/\Delta}$ mice treated with +PBS or +anti-LTβr antibody +LTα1/β2. Each circle represents the total numbers of HEVs/LN section or the total PNAd+ area/LN section [µm²] extracted from the micrographs (B, C), bars indicate

*Figure 4 continued on next page*

*Figure 4 continued*

the mean. Scale bars, 200 µm (**B, C**). \*\*\*p<0.0005 (two-tailed unpaired Student's *t*-test (**D**)). Data are shown for representative sections from 2x analyzed iLNs per mouse (**B, C**) selected from 21x – 33x individually analyzed sections of six mice per group (**D**).
DOI: https://doi.org/10.7554/eLife.41239.009

*supplement 1 (C-D)*). Taken together, these results suggest that HEV-DC interactions are dependent on S1PR1- but not S1PR3 signalling either in DCs or in high-endothelial cells.

## S1PR1-G$_i$ signalling, survival of high-endothelial cells, and CCL21 release from PNAd$^+$ HEVs in pLNs

Given that stimulation of S1PR3, rather than S1PR1, on mature BMDCs has been reported to control mature BMDC migration (*Maeda et al., 2007*), and, given our observation that a S1P-S1PR1 dependent survival of high-endothelial cells controls DC localization around HEVs, we further investigated the consequences of S1PR1-G$_i$ signalling in HEVs. Immunohistochemistry and FACS revealed S1PR1 expression on PNAd$^+$ HEVs in pLNs of *Spns2*$^{f/f}$ and *Lyve1;Spns2*$^{\Delta/\Delta}$ mice (*Figure 6 (A–B)*). We assayed PNAd$^+$ HEVs in pLN-sections of *Spns2*$^{f/f}$ and *Lyve1;Spns2*$^{\Delta/\Delta}$ mice for phosphorylated Akt (pAkt) which can be induced by several upstream signalling axes including S1PR1-G$_i$ signalling (*Ishii et al., 2004*). PNAd$^+$ HEVs in pLNs of *Spns2*$^{f/f}$ showed distinct phosphorylation of Akt in contrast to the reduced levels of pAkt in high-endothelial cells in pLNs of *Lyve1;Spns2*$^{\Delta/\Delta}$ mice (*Figure 5—figure supplement 1 (E)*). Hence, it is possible that a S1PR1-G$_i$ directed signalling pathway in HEVs of pLNs from *Lyve1;Spns2*$^{\Delta/\Delta}$ mice appears to be defective, even though the proportion of S1PR1 surface expression was mildly increased in comparison to those in HEVs of pLNs from *Spns2*$^{f/f}$ (*Figure 6 (B)*).

The results of the S1PR-antagonist experiments and the S1PR1 expression analyses guided us to conditionally delete *S1pr1* on high-endothelial cells and to analyse if DCs are able to co-localize with *S1pr1*-deficient high-endothelial cells in pLNs. Therefore, we opted to generate *Cdh5*$^{CRE-ERT2}$; *S1pr1*$^{\Delta/\Delta}$ mice. Flow-cytometry revealed the efficient deletion of S1pr1 in HEVs and LECs in the conditional *Cdh5*$^{CRE-ERT2}$;*S1pr1*$^{\Delta/\Delta}$ mice by postnatal tamoxifen administration (*Figure 6—figure supplement 1 (A-B)*). *Cdh5*$^{CRE-ERT2}$;*S1pr1*$^{\Delta/\Delta}$ mice did not show any differences in CD4 or CD8 single-positive T-cell and mature rec. B-cell numbers in peripheral blood or pLNs (*Figure 6—figure supplement 1 (C-D)*). Strikingly, in these mice, mature wildtype BMDCs fail to co-localize with *S1pr1*-deficient HEVs, as also observed in *Lyve1;Spns2*$^{\Delta/\Delta}$ mice or WT mice treated with S1PR1-antagonists. Indeed, when we performed WT BMDC injections into the footpad of *Cdh5*$^{CRE-ERT2}$;*S1pr1*$^{\Delta/\Delta}$ mice (*Figure 6 (C)*) and analysed HEV-DC interactions 24 hr later, we could detect a remarkable impairment of total DC-numbers within a restricted area of 0–60 µm around the basal lamina of HEVs (*Figure 6 (D–F)*), while no differences in DC-positioning close to cortical lymphatics were observed (*Figure 6—figure supplement 1 (E-G)*). These results demonstrate that S1PR1-G$_i$ signalling on high-endothelial cells rather than on DCs is responsible for the impaired HEV-DC interaction.

This raises the question if the autocrine activation of S1PR1-G$_i$ signalling that controls survival of high-endothelial cells concomitantly influences chemotactic recruitment of DCs to HEVs. CCR7 and its ligand CCL21 are essentially involved in the extravasation of T- and B-cells through lymph node HEVs and in the homing of various subpopulations of mature antigen-presenting DCs through afferent lymphatics to the lymph nodes (*Martln-Fontecha et al., 2003*; *Ohl et al., 2004*). Importantly, CCL21 is produced by the majority of HEVs in murine pLNs (*Yang et al., 2007*; *Gunn et al., 1998*). Hence, we analysed CCL21 expression of PNAd$^+$ HEVs in pLN sections. We found a drastic reduction of the CCL21 signal in a restricted area around HEVs within a distance of 40 µm from the basal lamina of HEVs in pLNs from *Lyve1;Spns2*$^{\Delta/\Delta}$ mice when compared to those of the controls (*Figure 7 (A–B)*). Quantitative RT-PCR revealed that expression levels of CCL19 and CCL21, but not CXCL13, were significantly reduced in CD45$^-$/CD31$^+$/PNAd$^+$ high-endothelial cells isolated from pLNs of *Lyve1;Spns2*$^{\Delta/\Delta}$ mice when compared to those of *Spns2*$^{f/f}$ mice (*Figure 7 (C)*). In addition, high-endothelial cells of pLNs from wildtype mice treated with the S1PR1-specific antagonists FTY720 or W146 show a significant reduction of CCL21 release, independently of treatment with exogenous S1P, in vitro (*Figure 7 (D)*). These data indicate a dependency of CCL21 expression and release on the

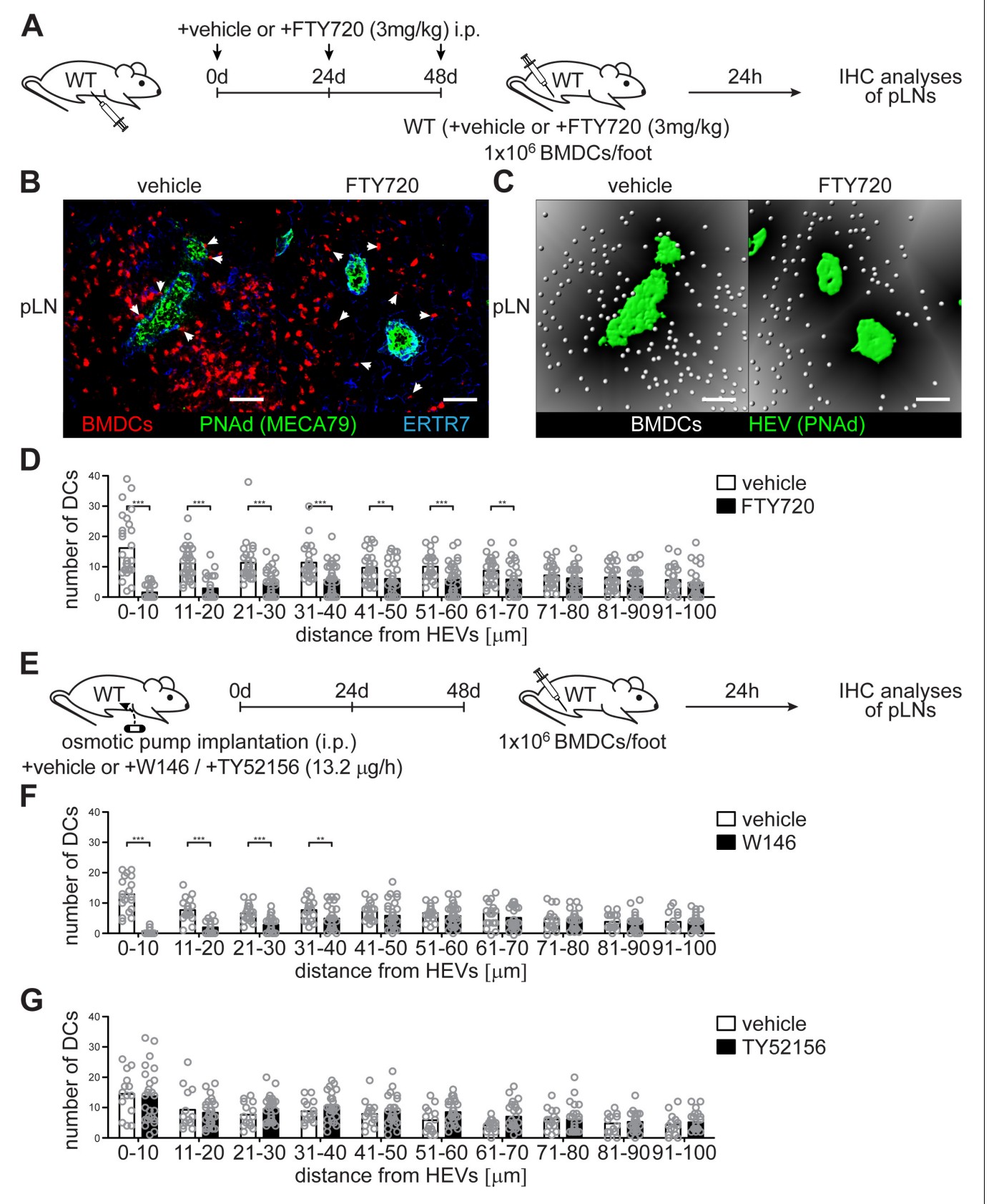

**Figure 5.** Co-localization of PNAd⁺ HEVs with lymph-derived BMDCs in pLNs is dependent on S1PR1- but not S1PR3-signalling. (**A**) Experimental flow-chart for the administration of the non-specific S1PR-antagonist FTY720 *i.p.* and lymphatic homing assays of footpad injected BMDCs to quantify HEV-DC interactions in pLNs in situ. (**B**) Confocal microscopy of pLNs of vehicle (left) or FTY720 (right) treated mice for CMTMR⁺ BMDCs (red), PNAd⁺ (green) HEVs and ERTR7⁺ (blue) fibroblastic tissue networks. (**C**) Visualisation of the distance of individual CMTMR⁺ BMDCs (white spheres) from PNAd⁺ HEVs (green surface) in pLNs of vehicle (left) or FTY720 (right) treated mice. Grey gradients visualise the distance transformation from HEVs (green surface) defined by PNAd-staining. (**D**) Total numbers of BMDCs (white spheres in (**B**)) in distances from 0 µm - 100 µm from HEVs (green surface in (**B**)) counted in 10 µm radial areas around HEVs in pLNs of vehicle or FTY720 treated mice. (**E**) Experimental flow-chart for the administration of the specific S1PR1-antagonist W146 and the S1PR3-antagonist TY52156, and lymphatic homing assays of BMDCs to quantify HEV-DC interactions in pLNs in situ. (**F**, **G**) Total numbers of BMDCs (white spheres as shown in (**C**)) in distances from 0 µm - 100 µm from HEVs counted in 10 µm radial areas around HEVs in pLNs of treated mice. Each circle represents the total numbers of BMDCs around HEVs in the visual field of a micrograph (**D**, **F**, **G**); bars indicate the mean. Scale bars, 50 µm (**B**, **C**). **p<0.005; ***p<0.0005 (two-tailed unpaired Student's *t*-test (**F**, **G**)). Data are representative for 37x representative individual sections of 2x analyzed popliteal LNs per mouse pooled from two independent experiments (**B**, **C**, **D**) with n = 6 mice per group (**B**, **C**, **D**) and for 34x (**F**) or 26x (**G**) representative individual sections of 2x analyzed popliteal LNs per mouse pooled from 5x mice per group (**F**, **G**).

DOI: https://doi.org/10.7554/eLife.41239.010

The following figure supplement is available for figure 5:

**Figure supplement 1.** Co-localization of lymph-derived BMDCs with PNAd⁺ HEVs in pLNs is dependent on S1PR1- but not S1PR3-signaling.

DOI: https://doi.org/10.7554/eLife.41239.011

---

integrity of high-endothelial cells, which is warranted by the autocrine activated S1P/S1PR1-$G_i$ signalling axis, and, initialized by SPNS2-dependent S1P-secretion from HEVs.

## Discussion

In the present study we have demonstrated that chemotactic recruitment of lymph-derived DCs to HEVs is controlled by a SPNS2-dependent S1P release from HEVs, and autocrine S1PR1-$G_i$ signalling on high-endothelial cells, which collectively warrants survival of HEVs. We followed the development of hypotrophic pLNs, which occurs as a consequence of severely impaired HEV-morphology and -function in *Lyve1;Spns2*$^{\Delta/\Delta}$ mice. The usually cuboidal morphology of HEVs has been considered to be a consequence of lymphocyte accumulation in the cytoplasm of high-endothelial cells, but electron microscopic analyses have revealed that when mice were depleted of circulating lymphocytes the appearance of high-endothelial cells remains to be unaltered (*Yamaguchi and Schoefl, 1983*; *Schoefl, 1972*). These observations are in line with our previous studies, in which we have isolated high-endothelial cells for cultivation. In these studies, consistent with their in vivo morphology, isolated HEVs retained invariably large and plump in the absence of lymphocytes, having voluminous cytoplasm (*Matsutani et al., 2007*; *Bai et al., 2013*). Therefore, the impairment of the high-endothelial cell morphology in *Spns2*-deficient HEVs appears to be independent of the impaired transcellular immigration of lymphocytes into pLNs. We have chosen the *Lyve1*$^{CRE}$ mouse model in order to delete *Spns2* in LECs and HEVs, since the HEV-specific *GlcNAc6ST-2*$^{-CRE}$ mouse line (*Kawashima et al., 2009*), is, unfortunately, no longer available (personal communication, Dr. Hiroto Kawashima to Dr. Masaru Ishii). As a result of altered HEV-integrity in pLNs of *Lyve1;Spns2*$^{\Delta/\Delta}$ mice we hypothesized that fetal progenitors of high-endothelial cells express LYVE1 during embryogenesis. This hypothesis is supported by tdTOMATO expression in PNAd⁺ HEVs to similar levels as of LECs of pLNs of WT *Lyve1;tdTomato* mice. In accordance with this observation Pham et al. have described CRE-recombinase activity in subsets of BECs of *Lyve1*$^{CRE}$ mice, while the minority of lymph node FRCs (<7% of all LN FRCs) showed CRE-recombinase activity when mice were intercrossed to mice carrying YFP preceded by a floxed transcriptional stop in the Rosa26 locus (*Pham et al., 2010*). The relatively small total number of HEVs compared to other BECs is considered unlikely to account for a significant contribution to S1P levels in peripheral blood. Hence, we detected unaltered S1P-levels in the blood of *Lyve1;Spns2*$^{\Delta/\Delta}$ mice. Our immunostainings confirmed LYVE1 expression in developing high-endothelial cell progenitors and HEVs of iLNs of wildtype embryos of E16.5 and E18.5. In line with this observation, the heterogeneity in LYVE1 expression has been discussed to represent distinction in the phenotype of individual arterial and venous endothelial cells in the embryo (*Gordon et al., 2008*). Moreover, our results reveal that *Spns2* transcripts in purified high-endothelial cells from *Lyve1;Spns2*$^{\Delta/\Delta}$ mice were significantly reduced. Besides LECs, subsets of hematopoietic cells, for example resident macrophages, in the pLNs, have been suggested to express LYVE1. This raises the possibility that CRE recombination in *Lyve1*$^{CRE}$ mice could occur in

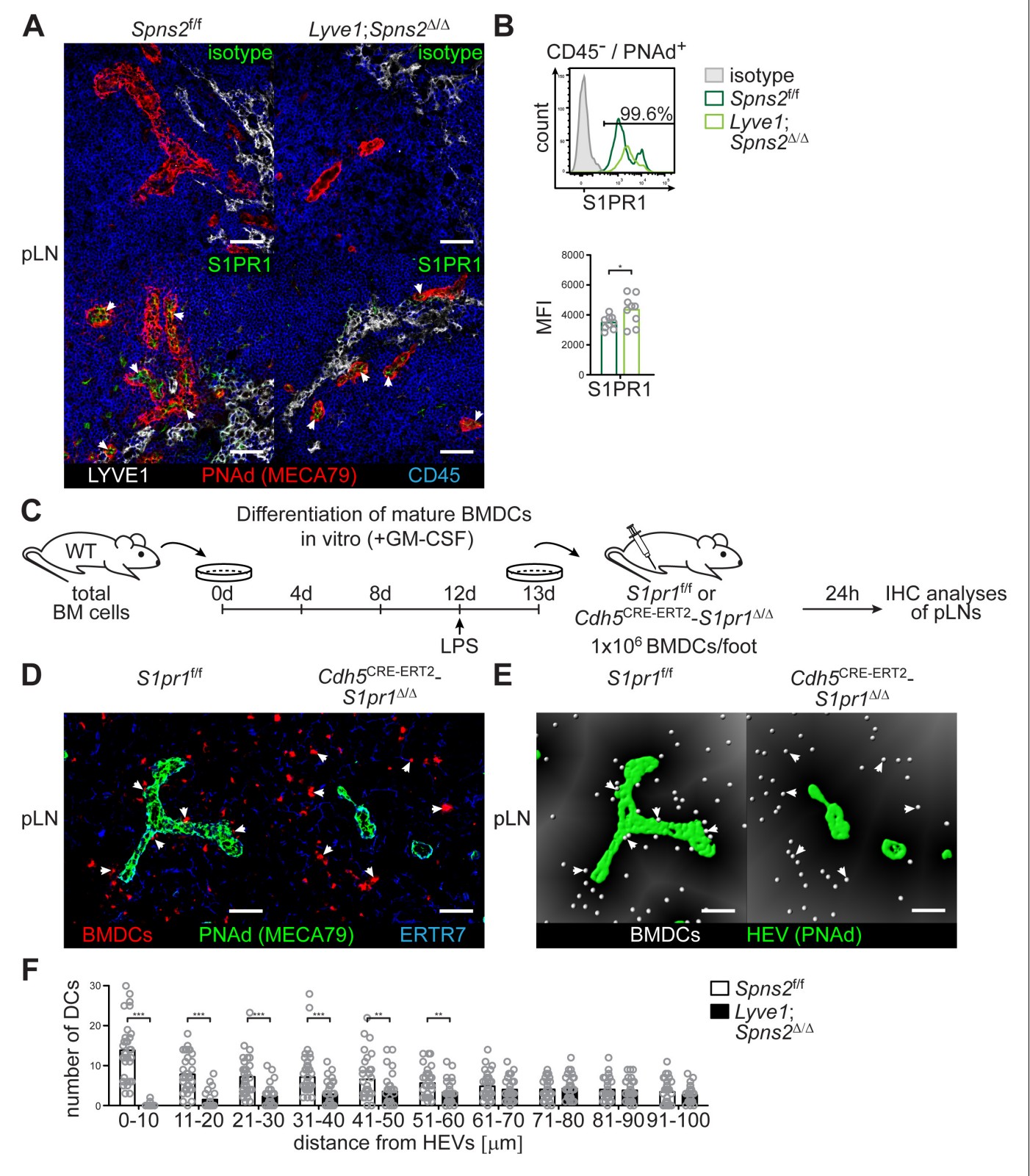

**Figure 6.** SPNS2-derived S1P controls autocrine S1PR1-G$_i$ signalling in PNAd$^+$ HEVs of pLNs. (**A**) Fluorescence microscopy of pLNs of *Spns2*$^{f/f}$ (left) and *Lyve1;Spns2*$^{\Delta/\Delta}$ (right) mice for isotype (top, green)/S1PR1 (bottom, green) on PNAd$^+$ (red) HEVs and LYVE1$^+$ (white) LECs, and CD45$^+$ (blue) hematopoietic cells. (**B**) FACS analysis of the cell surface expression of S1PR1 on CD45$^-$/CD31$^+$/PNAd$^+$ high-endothelial cells in pLNs of *Spns2*$^{f/f}$ and *Lyve1;Spns2*$^{\Delta/\Delta}$ mice. (**C**) Experimental flow-chart of BMDC-differentiation in vitro, and lymphatic homing assays of footpad injected BMDCs to quantify

*Figure 6 continued on next page*

*Figure 6 continued*

DC-immigration from afferent lymphatics into pLNs of *S1pr1*^f/f and *Cdh5*^CRE-ERT2;*S1pr1*^Δ/Δ mice. (D) Confocal microscopy of pLNs of *S1pr1*^f/f (left) and *Cdh5*^CRE-ERT2;*S1pr1*^Δ/Δ (right) mice for CMTMR^+ BMDCs (red), PNAd^+ (green) HEVs and ERTR7^+ (blue) fibroblastic tissue networks. (E) Visualisation of the automated detection of individual CMTMR^+ BMDCs (white spheres) from PNAd^+ HEVs (green surface) in pLNs of *S1pr1*^f/f (left) and *Cdh5*^CRE-ERT2; *S1pr1*^Δ/Δ (right) mice. Grey gradients visualise the distance transformation from HEVs (green surface) defined by PNAd-staining. (F) Total numbers of BMDCs (white spheres in (E)) in distances from 0 µm - 100 µm from HEVs counted in 10 µm radial areas around HEVs in pLNs of *S1pr1*^f/f and *Cdh5*^CRE-ERT2;*S1pr1*^Δ/Δ mice. Each circle represents an individual mouse (B) or total numbers of BMDCs around HEVs in the visual field of a micrograph (F); bars indicate the mean. Scale bars, 50 µm (A, D, G). *p<0.05, **p<0.005; ***p<0.0005 (two-tailed unpaired Student's *t*-test (B, F)). Data are representative for five mice per group pooled from two independent experiments (A) with n = 2 or n = 3 mice per group (A), or are representative for nine mice per group pooled from three independent experiments (B) with n = 3 mice per group (B), or for 37x representative individual sections of 2x analyzed popliteal LNs per mouse pooled from two independent experiments (F) with n = 4 mice per group (H).

DOI: https://doi.org/10.7554/eLife.41239.012

The following figure supplement is available for figure 6:

**Figure supplement 1.** Endothelial-cell specific deletion of *S1pr1* does not affect lymphocyte immigration into pLNs, and does not influence DC-positioning at cortical lymphatics.

DOI: https://doi.org/10.7554/eLife.41239.013

hematopoietic cells in pLNs of *Lyve1;Spns2*^Δ/Δ and, thus, alterations of S1P-release from hematopoietic cells may also influence high-endothelial cell integrity. However, bone marrow chimera experiments reconstituting CD45.2^+ *Lyve1*^CRE;*Sphk*-deficient mice with CD45.1^+ wild-type BM showed no significant alterations in S1P concentration (*Pham et al., 2010*). In addition, neither immature RBCs in the blood, nor CD4^+ or CD8^+ or CD19^+ cells express SPNS2 in the lymph node environment (*Mendoza et al., 2012*). In summary, we conclude that *Spns2*-deletion during ontogeny and impaired S1P release from high-endothelial cells is responsible for the impaired development of HEVs in pLNs of adult *Lyve1;Spns2*^Δ/Δ mice.

S1P in the blood has been described as a regulator of vascular development (*Xiong and Hla, 2014*; *Xiong et al., 2014*). Furthermore, HDL/ApoM-associated S1P has been accredited for positively influencing vascular integrity (*Xiong and Hla, 2014*). We observed a reduced S1PR1-G$_i$ signalling represented by reduced pAkt levels in high-endothelial cells in pLNs of *Lyve1;Spns2*^Δ/Δ mice. Furthermore, we monitored a strong reduction of S1P secretion from LECs in those animals. However, we consider the low nanomolar levels of S1P in the abluminal sites of pLNs (*Pappu et al., 2007*; *Schwab, 2005*) and significant S1P-lyase activity in hematopoietic cells (*Schwab, 2005*) of the LNs as indications that make it unlikely that LECs derived S1P, or S1P secreted from the minor SPNS2 expressing FRC populations of the pLNs, activates S1PR1-G$_i$ signalling in HEVs. Thus, we conclude that the reduced S1PR1-G$_i$ signalling is caused by the impaired autocrine activation through HEV-derived S1P. Interestingly, pLNs of immunized mice lacking S1P in the plasma (*Camerer et al., 2009*) exhibit impaired HEV integrity similar to podoplanin and Clec2-deficient mice (*Herzog et al., 2013*). While platelet-derived S1P is considered to directly influence HEV-barrier function by promoting endothelial adherens junctions through upregulation of VE-cadherin on HEVs (*Herzog et al., 2013*), we provide evidence that S1P produced and secreted by SPNS2 from HEVs regulates self-production and release of CCL21 by affecting the viability of high-endothelial cells. An alternative scenario may be that disruption of S1P production releases the inhibition of a chemo-repellent molecule downstream of S1PR1-G$_i$ signalling that prevents migration of DCs towards HEVs. However, we observed a S1PR1-G$_i$ signalling dependent regulation of the DC chemo-attractant CCL21 which makes the chemo-repellent phenomenon less likely. Depletion of DCs in vivo has been implicated in the reversion of the adult and 'peripheral' PNAd^+/MAdCAM-1^- HEV phenotype to a neonatal or 'mucosal' PNAd^-/MAdCAM-1^+ phenotype resulting in inhibited lymphocyte recruitment and the development of hypotrophic pLNs (*Moussion and Girard, 2011*). Additionally, CCR7-expressing DCs were described to directly contribute to HEV growth by promoting the production of VEGF from FRCs (*Wendland et al., 2011*; *Chyou et al., 2011*; *Kumar et al., 2015*). These reports are in line with our observation of impaired chemotactic recruitment of transplanted BMDCs to HEVs, and therefore reduced HEV-DC interactions in pLNs of *Lyve1;Spns2*^Δ/Δ mice. Importantly, positioning of transplanted BMDCs around cortical lymphatics in pLNs of *Lyve1;Spns2*^Δ/Δ mice in comparison to *Spns2*^f/f mice was unaltered, underlining the high-endothelial specificity of this phenomenon. Indeed, the extensive DC-devoid area around HEVs in pLNs of *Lyve1;Spns2*^Δ/Δ mice points toward an

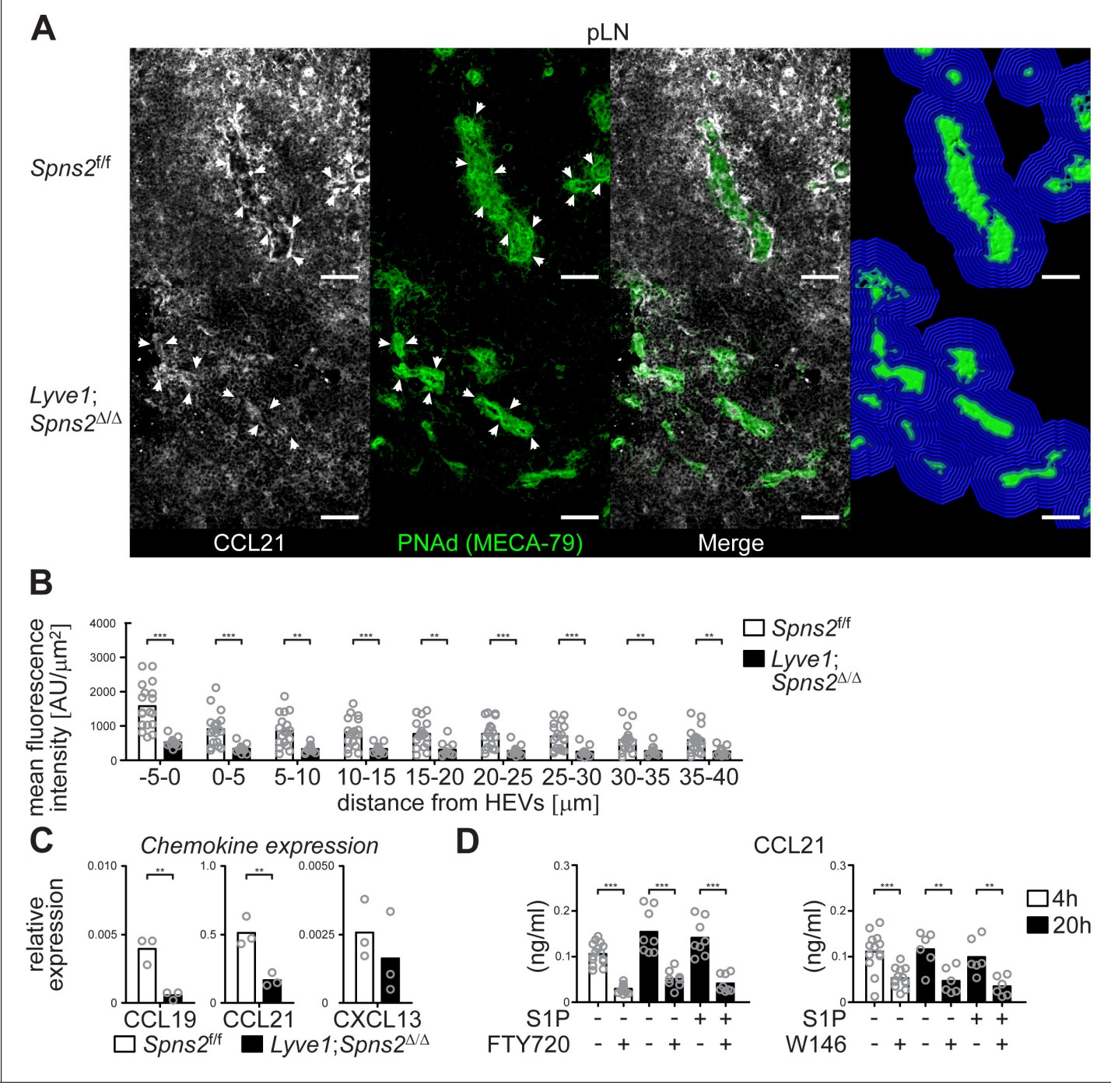

**Figure 7.** CCL21-production and -release from HEVs is severely impaired in pLNs of *Lyve1;Spns2*$^{\Delta/\Delta}$ mice. (**A**) The IHC analysis of CCL21 (white) distribution around PNAd$^+$ HEVs, and visualisation of the automated detection of PNAd$^+$ HEVs (green surface) and of the radial areas (blue) around HEVs used for the quantification of the mean fluorescent intensity of the CCL21 signal in pLNs of in *Spns2*$^{f/f}$ (top) and *Lyve1;Spns2*$^{\Delta/\Delta}$ (bottom) mice. (**B**) The mean fluorescent intensity of the CCL21 signal in distances from −5 µm to 40 µm from the outer border of HEVs (green surface in (**C**)) determined in 5 µm radial areas around HEVs in pLNs of *Spns2*$^{f/f}$ and *Lyve1;Spns2*$^{\Delta/\Delta}$ mice. (**C**) qRT-PCR analysis of CCL19, CCL21 and CXCL13 expression levels in total mRNA isolated from CD45$^-$/CD31$^+$/PNAd$^+$ high-endothelial cells sorted from pLNs of *Spns2*$^{f/f}$ and *Lyve1;Spns2*$^{\Delta/\Delta}$ mice. (**D**) ELISA of the CCL21 levels of the supernatant of high-endothelial cells cultivated with or without 10 µM FTY720 (left) or 10 µM W146 (right) in vitro. Each circle represents the mean fluorescent intensity of the CCL21 signal detected around HEVs in the visual field of a micrograph (**B**), the relative chemokine expression levels in mRNA extracted from the total CD45$^-$/CD31$^+$/PNAd$^+$ high-endothelial cells (**C**), or the CCL21 protein levels detected in the supernatant of individual cell cultures (**D**) of CD45$^-$/CD31$^+$/PNAd$^+$ high-endothelial cells; bars indicate the mean. Scale bars, 50 µm (**A**). **p<0.005; ***p<0.0005 (two-tailed unpaired Student's *t*-test (**B–D**)). Data are representative for 18x individual sections of 2x analyzed pLNs, iLNs and bLNs per mouse pooled from three

*Figure 7 continued on next page*

*Figure 7 continued*

mice per group (**A**, **B**), three independent mRNA preparations of 2x pLNs, iLNs and bLNs per mouse pooled from five mice per group (**C**), or three independent stimulations with n = 2 to n = 4 of a total of 8x – 16x (FTY720) or 6x – 12x (W146) individual cell cultures (**D**) with total sorted high-endothelial cells from 2x pLNs, iLNs and bLNs per mouse pooled from five mice per group in vitro.

DOI: https://doi.org/10.7554/eLife.41239.014

involvement of perivascular/peri-HEV stromal in the development of the observed phenotype. Concomitant with this idea DC-derived lymphotoxin beta receptor LTβR ligands have been described as critical mediators of reticular cell survival by modulating podoplanin (*Kumar et al., 2015*; *Chyou et al., 2011*). Therefore, it is likely that reduced chemotactic recruitment of DCs to HEVs results in reduced podoplanin modulated and integrin-mediated DC-FRC adhesion (*Kumar et al., 2015*), which maintains survival in peri-HEV stromal cells. We could partially rescue the total numbers and the morphology of HEVs in pLNs of *Lyve1;Spns2*$^{\Delta/\Delta}$ mice by administration of agonistic anti-LTβR antibody and recombinant LTα1β2 protein which, again, indicates that DC-derived LTα1β2 is an integral part of the factors facilitating HEV-integrity and -function in vivo, and, thus, lymphocyte immigration to lymph nodes (*Moussion and Girard, 2011*; *Browning et al., 2005*).

We have observed that *Spns2*-deficient high-endothelial cells are highly apoptotic in comparison to wildtype controls. However, in our analyses the frequencies of apoptotic lymphocytes in the lymph nodes of *Lyve1;Spns2*$^{\Delta/\Delta}$ mice is decreased in comparison to *Spns2*$^{f/f}$ controls. Mendoza et al. have recently reported that lymph node T-cells of *Lyve1;Spns2*$^{\Delta/\Delta}$ mice show mitochondrial dysfunction due to reduced S1PR1-G$_i$ signalling, and, as a consequence, appear to be apoptotic (*Mendoza et al., 2017*). We therefore imagine that the high phagocytic activity in relation to the strongly reduced lymphocytes numbers is possibly responsible for a very efficient clearance of any apoptotic cells in the LN-microenvironment of *Lyve1;Spns2*$^{\Delta/\Delta}$ mice, and, in fact, reduces the frequencies of apoptotic B- or T-cells in those mice in comparison to the controls. We consider it likely that the regulation of cell survival by S1PR1-G$_i$ signalling as observed by Mendoza et al. is a cellular mechanism that also could account for the strongly increased apoptosis in high-endothelial cells we detected in pLNs of *Lyve1;Spns2*$^{\Delta/\Delta}$ mice. However, we assume that this mechanism on high-endothelial cells is not solely responsible for high-endothelial survival since we were able to significantly rescue the total HEV numbers and their morphology by treatment with an agonistic anti-LTβR antibody and recombinant LTα1β2 protein.

Nevertheless, our data acquired from the BMDC transfer experiments in *Cdh5*$^{CRE-ERT2}$;*S1pr1*$^{\Delta/\Delta}$ mice strengthen the idea that S1PR1-Gi signalling on HEVs, and not on DCs, is responsible for the impaired co-localization of both cell types. In addition, our experimental setup excludes the idea that the impaired HEV-DC interactions are caused by secondary effects of impaired S1PR1-G$_i$ signalling on hematopoietic cells or by developmental defects on high-endothelial progenitors induced through reduced S1P-levels in the lymph node environment during ontogeny.

We firmly established a role of SPNS2-secreted S1P in autocrine CCL21-mediated regulation of HEV-DC interactions by deleting *S1pr1* on all vascular ECs in *Cdh5*$^{CRE-ERT2}$;*S1pr1*$^{\Delta/\Delta}$ or by inhibiting S1PR-signalling in HEVs through the administration of S1PR-antagonists. Interestingly, increasing expression of CCL21 has been described on FRCs and HEVs in the presence of DCs (*Wendland et al., 2011*). How the autocrine S1P/S1PR1-G$_i$ signalling is able to influence CCL21 expression, and possibly secretion, in HEVs is still a point of consideration. S1PR1-G$_i$ signalling controls the production of basic energy currency by oxidative phosphorylation and, therefore, promotes survival of naïve T-cells in pLNs (*Mendoza et al., 2017*). It is well known that high-endothelial cells show high metabolic activity and unlike normal ECs HEVs exhibit abundant numbers of mitochondria associated with the rough endoplasmic reticulum, a prominent Golgi complex, and large polyribosome clusters observation (*Girard et al., 1999*), an observation that we can confirm from our own experience. However, it appears that arterial, venous, lymphatic, and microvascular ECs use glycolysis as the predominant bioenergetic pathway (*De Bock et al., 2013*). Therefore, we can think of a scenario in which S1PR1-G$_i$ signalling is required to maintain mitochondrial content in high-endothelial cells reminiscent of recent observations made in naïve T-cells in which mitochondrial loss was observed as a consequence of increased mitophagy in *S1pr1*-deficient T-cells (*Mendoza et al.,*

*2017*). The reduced energy levels may impair CCL21 secretion and, therefore, reduced HEV-DC interactions triggering LTβr-signalling and survival of high-endothelial cells.

The negative regulation of CCL21 expression in high-endothelial cells by an impaired S1PR1-$G_i$ signalling throws light on the impaired transendothelial migration and the development of hypotrophic pLNs in *Lyve1;Spns2*$^{\Delta/\Delta}$ mice. The impaired interactions of DCs with high-endothelial cells and the concomitant impaired architecture of HEVs we have described to be one reason responsible for reduced lymphocyte immigration from the blood into the pLNs. However, CCL21 displayed by HEVs support T-cell arrest through integrin activation on high-endothelial cells which is impaired in the *plt/plt* (plt; paucity of lymph-node T cells) mutant mouse strain and results in greatly reduced T-cell entry into LN (*Gunn et al., 1998*; *Stein et al., 2000*; *Luther et al., 2000*). B-cells upregulate integrins in response to CXCL12 and CXCL13 in order to firmly arrest on HEVs. Both chemokines are produced abluminal by fibroblastic stromal cells and are subsequently transported by the fibroblastic conduit system to HEVs. Immobilization of the chemokines by heparan sulfate upon transcystosis through HEVs on the luminal cell surface ensures B cell recruitment (*Bao et al., 2010*). Therefore, the functional destruction of HEVs in pLNs of *Lyve1;Spns2*$^{\Delta/\Delta}$ mice, which display strong impairment of CCL21 expression and are possibly unable to shuttle CXCL12 and CXCL13 to their luminal surfaces, contributes to the severely impaired lymphocyte immigration causing the hypotrophy. Interestingly, impaired T-cell homeostasis in lymph nodes of conventional *Spns2*-KO and endothelial-cell specific *Tie2;Spns2*$^{\Delta/\Delta}$ mice has been described previously (*Fukuhara et al., 2012*; *Mendoza et al., 2012*; *Nagahashi et al., 2013*). In these studies the reduced lymphocyte numbers were reasoned as a consequence of lymphopenia in blood circulation induced through trapping of egress potent cells in the thymus (*Fukuhara et al., 2012*; *Mendoza et al., 2012*; *Nagahashi et al., 2013*). However, our observations imply that impairment of HEVs may additionally account for reduced lymphocyte numbers in pLNs of conventional *Spns2*-KO and endothelial-cell specific *Tie2;Spns2*$^{\Delta/\Delta}$ mice. Nevertheless, although we found impaired HEV-DC co-localization in *Cdh5*$^{CRE-ERT2}$;*S1pr1*$^{\Delta/\Delta}$ mice, lymphocyte numbers were not affected in pLNs of these mice. This observation is in line with previous descriptions of lymphocyte numbers in Cdh5$^{CRE-ERT2}$;S1PR1$^{\Delta/\Delta}$ mice (*Blaho et al., 2015*; *Galvani et al., 2015*). Therefore, we conclude that autocrine activation of S1PR1-signalling on HEV is mainly important for DC-localization and to a much lesser extent for the regulation of lymphocyte immigration from the blood circulation into pLNs.

Furthermore, we revealed that *Spns2* expressed in LECs is fundamentally important for maintaining S1P, but not other lysophospholipid levels in the lymph. Hence, the S1P-gradient from LNs towards the lymphatic system is disrupted in *Lyve1;Spns2*$^{\Delta/\Delta}$, which severely impairs the egress potential of recirculating lymphocytes from the LNs into the lymphatic vasculature. The strong reduction of S1P in lymph, but not blood, of *Lyve1;Spns2*$^{\Delta/\Delta}$ mice resembles S1P-levels in the circulatory fluids of *Lyve1;Sphk1*$^{\Delta/\Delta}$ mice (*Pham et al., 2010*). Given the, probably non-directional, S1P-secretion by LECs into the lumen of the lymphatics and the abluminal sides of the lymphoid organ, the low tonic S1P-concentrations in the interstitial sites of the LNs can be explained by S1P-lyase expression in the hematopoietic compartment that essentially degrades S1P (*Schwab, 2005*). Additionally, a role for the lipid phosphate phosphatase LPP3 has been identified in maintaining low tonic S1P concentrations in the thymus and spleen (*Bréart et al., 2011*; *Ramos-Perez et al., 2015*) and it can be speculated that LPP3 also accounts for low interstitial S1P concentrations in the LNs.

A constantly growing body of work indicates that S1P is important to maintain vascular integrity and regulate vascular leak. Mice lacking both spingosine-kinases ('pS1Pless' mice) show increased vascular leak at basal and inflammatory conditions (*Camerer et al., 2009*). Indeed, circulating S1P that binds to high-density lipoprotein via its carrier apolipoprotein M (ApoM) and mediates S1P-dependent protection of the endothelial barrier by stimulating S1PR1 signalling (*Argraves et al., 2011*; *Christensen et al., 2016*; *Christoffersen et al., 2011*). It has been reported that conventional *Spns2*-KO mice exhibit reduced lymphatic network formation in lymph nodes (*Nagahashi et al., 2013*), enhancing the idea that SPNS2-secreted circulating S1P is important for maintaining vascular integrity. However, in *Lyve1;Spns2*$^{\Delta/\Delta}$ mice it appears that plasma S1P levels are not affected, while S1P levels in the lymphatics are strongly reduced. Therefore, the already relatively low concentrations of S1P in the lymph node parenchyma are even more affected. This raises the question how high-endothelial cells sense and respond to SPNS2-derived S1P with a concomitant autocrine S1PR-signalling in *Lyve1;Spns2*$^{\Delta/\Delta}$ mice, because apically high-endothelial cells are exposed to high S1P concentrations, while on the basolateral side S1P concentrations are low. Our model proposes that

high-endothelial cells polarize in their responsiveness to an autocrine activation pathway. We believe that SPNS2-derived S1P from high-endothelial cells is able to stimulate CCL21 release from HEVs, which, in turn, facilitates vascular-integrity promoting HEV-DC interactions. How low tonic S1P-signals provided basolateral in the lymph node parenchyma do stimulate S1PR-signalling on HEVs, while high S1P concentrations in blood plasma are unable to cause the same effect are points of consideration. It recently has been shown that lysophosphatidic acid (LPA) receptor-1 (LPAR1) suppresses cell-surface $S1PR1/G\alpha_i$ signalling on LECs by inter-G protein–coupled receptor β-arrestin coupling (*Hisano et al., 2019*). Interestingly, we and others could show that HEVs express high-levels of autotaxin, an ectoenzyme that catalyzes the conversion of lysophosphatidylcholine (LPC) to LPA (*Kanda et al., 2008*; *Nakasaki et al., 2008*). Furthermore, we could show that activation of autotaxin is LTβR signalling dependent (*Takeda et al., 2016*). However, it needs more in depth analyses in order to clarify if a potential involvement of LPAR1-dependent suppression of $S1PR1/G\alpha_i$ signalling may also play a role in high-endothelial cells.

MZ B cells have been shown to gather blood-borne antigens in the splenic marginal zone. Localization and retention of MZ B cells in the splenic marginal zone has been accredited respectively to S1PR1 expressed on MZ B cells and to its ligand S1P, which is provided in high concentrations by constant blood flow through the open structures of the marginal sinuses at the border between the red and white pulp of the spleen (*Cinamon et al., 2004*; *Cinamon et al., 2008*). Moreover, a fine balance in signalling activities of MZ B cell expressed S1PR1 and CXCR5, the receptor for the B lymphocyte chemoattractant CXCL13, facilitates the continuous marginal zone-follicular shuttling of MZ B cells (*Cinamon et al., 2004*; *Cinamon et al., 2008*). We observed a significant increase of MZ B cells whereas FO B cells are decreased in *Lyve1;Spns2*$^{\Delta/\Delta}$ mice reminiscent of observations made with *S1pr1*-deficient B cells transferred into CXCL13-deficient host (*Cinamon et al., 2004*). In the present study we were unable to detect alterations in plasma S1P concentrations of *Lyve1;Spns2*$^{\Delta/\Delta}$ mice supporting the assumption that MZ B cell accumulation is mediated by a S1P/S1PR1-retention signal. This observation is in line with experiments that indicate that S1P-mediated signalling dominates the attraction along CXCL13 cues into the follicle (*Cinamon et al., 2004*; *Cinamon et al., 2008*). Therefore, we propose that CXCR5/CXCL13-dependent recruitment of MZ B cells into the follicle is impaired in the spleen of *Lyve1;Spns2*$^{\Delta/\Delta}$ mice. This raises the question of which *Lyve1*-expressing cells may be responsible for a modulation of CXCL13 concentration in splenic follicles. *Lyve1*-expression is not restricted to the lymphatic vasculature and data presented in this study and by other groups show that LYVE1 is expressed on uncommitted embryonic blood vessels (*Gordon et al., 2008*). Recent findings show that LYVE1 is expressed on a subset of sinusoidal endothelial cells in the spleen of humans and adult rodents (*Banerji et al., 1999*; *Zheng et al., 2016*). LYVE1 is also expressed by resident macrophage populations and a subset of infiltrating macrophages found in tumors and inflamed tissues (*Jackson, 2004*), and splenic stromal organizer cells during ontogeny (*Tan and Watanabe, 2017*). Therefore, further investigations are necessary to clarify if either of these cell populations directly or indirectly affects cell populations that express CXCL13 in the splenic B cell follicles by S1P secreted from SPNS2.

In summary, our results reveal a previously unsuspected role of high-endothelial cell-derived S1P in maintaining HEV-integrity by facilitating HEV-DC interactions. These findings give new insights into the S1P-mediated autocrine regulation of $S1PR1-G_i$-signalling dependent survival of high-endothelial cells and CCL21-secretion of HEVs, as well as the regulation of lymphocyte recirculation during immune surveillance.

# Materials and methods

**Key resources table**

| Reagent type (species) or resource | Designation | Source or reference | Identifiers | Additional information |
|---|---|---|---|---|
| Strain, strain background (*Mus musculus*) | C57BL/6 (B6) mice | CLEA Japan, Inc, Tokyo | RRID: IMSR_JAX:000664 | |

*Continued on next page*

*Continued*

| Reagent type (species) or resource | Designation | Source or reference | Identifiers | Additional information |
|---|---|---|---|---|
| Strain, strain background (*Mus musculus*) | Lyve1CRE mice | *Pham et al., 2010* | RRID: IMSR_JAX:012601 | |
| Strain, strain background (*Mus musculus*) | Spns2f/f mice | *Fukuhara et al., 2012* | RRID: MGI:5426399 | |
| Strain, strain background (*Mus musculus*) | Ai9 mice | *Madisen et al., 2010* | RRID: Addgene_22799 | |
| Strain, strain background (*Mus musculus*) | Cdh5CRE-ERT2 mice | *Okabe et al., 2014* | | |
| Strain, strain background (*Mus musculus*) | S1pr1f/f | *Allende et al., 2003* | RRID: MGI:2681963 | . |
| Antibody | rat anti-mouse/human monoclonal PNAd (MECA-79) | *Streeter et al., 1988a* | | [2 µg/ml] |
| Antibody | rat anti-mouse monoclonal MAdCAM1 (MECA-367) | *Streeter et al., 1988b* | | [2 µg/ml] |
| Antibody | rat anti-mouse monoclonal CD11a (M17/4) | BioXcell, West Lebanon, NH, USA | RRID: AB_1107582 | [2 µg/ml] |
| Antibody | rat anti-mouse/human monoclonal CD49d (PS/2) | BioXcell, West Lebanon, NH, USA | RRID: AB_1107657 | [2 µg/ml] |
| Antibody | rabbit anti-mouse polyclonal collagen IV antibody (LB-1403) | Cosmo Bio Co., Ltd, Tokyo | | [2 µg/ml] |
| Antibody | rat anti-mouse monoclonal CD4 (RM4-5) | BD Biosciences, San Jose, CA | RRID: AB_393575 | [2 µg/ml] |
| Antibody | rat anti-mouse monoclonal IgA (C10-3) | BD Biosciences, San Jose, CA | RRID: AB_396541 | [2 µg/ml] |
| Antibody | rat anti-mouse monoclonal CCR7 (4B12) | BD Biosciences, San Jose, CA | | [2 µg/ml] |
| Antibody | rat anti-mouse monoclonal CD45 (30-F11) | eBioscience, Inc, San Jose, CA | RRID: AB_10373710 | [2 µg/ml] |
| Antibody | rat anti-mouse monoclonal CD8 (53–6.7) | eBioscience, Inc, San Jose, CA | RRID: AB_11155388 | [2 µg/ml] |

*Continued on next page*

*Continued*

| Reagent type (species) or resource | Designation | Source or reference | Identifiers | Additional information |
|---|---|---|---|---|
| Antibody | rat anti-mouse monoclonal IgM (II/41) | eBioscience, Inc, San Jose, CA | RRID: AB_467582 | [2 µg/ml] |
| Antibody | rat anti-mouse monoclonal IgD (11-26) | eBioscience, Inc, San Jose, CA | RRID: AB_465346 | [2 µg/ml] |
| Antibody | rat anti-mouse monoclonal CD23 (B3D4) | eBioscience, Inc, San Jose, CA | RRID: AB_466392 | [2 µg/ml] |
| Antibody | rat anti-mouse monoclonal LYVE1 (ALY7) | eBioscience, Inc, San Jose, CA | RRID: AB_1633414 | [2 µg/ml] |
| Antibody | rat anti-mouse monoclonal CD45.1 (A20) | BioLegend, San Diego, CA | RRID: AB_313491 | [2 µg/ml] |
| Antibody | hamster anti-mouse monoclonal CD3 (145–2 C11) | BioLegend, San Diego, CA | RRID: AB_312666 | [2 µg/ml] |
| Antibody | rat anti-mouse/human monoclonal B220 (RA3-6B2) | BioLegend, San Diego, CA | RRID: AB_312986 | [2 µg/ml] |
| Antibody | rat anti-mouse monoclonal CD21/35 (7E9) | BioLegend, San Diego, CA | RRID: AB_940411 | [2 µg/ml] |
| Antibody | rat anti-mouse monoclonal I-A/I-E (M5/114.15.2) | BioLegend, San Diego, CA | RRID: AB_313316 | [2 µg/ml] |
| Antibody | hamster anti-mouse monoclonal cD11c (N418) | BioLegend, San Diego, CA | RRID: AB_313770 | [2 µg/ml] |
| Antibody | rat anti-mouse monoclonal ERTR7 | Santa Cruz Biotechnology, Inc, Dallas, TX | | [2 µg/ml] |
| Antibody | rabbit anti-mouse polyclonal EDG-1 (H-60) | Santa Cruz Biotechnology, Inc, Dallas, TX | RRID: AB_2184743 | [2 µg/ml] |
| Antibody | rabbit anti-mouse polyclonal pAkt (Ser473) | Cell Signaling Technology, Danvers, MA | RRID: AB_329825 | [2 µg/ml] |
| Peptide, recombinant protein | rabbit IgG isotype control | Cell Signaling Technology, Danvers, MA | RRID: AB_1550038 | [2 µg/ml] |
| Antibody | rat anti-mouse monoclonal CD16/CD32 | BioLegend, San Diego, CA | RRID: AB_312800 | [2 µg/ml] |
| Antibody | rat anti-mouse monoclonal LTβr (5G11) | BioLegend, San Diego, CA | RRID: AB_1659177 | [2 µg/ml] |
| Antibody | goat anti-mouse polyclonal CCL21 (AF457) | R and D Systems Inc, Minneapolis, MN | RRID: AB_2072083 | [2 µg/ml] |
| Peptide, recombinant protein | recombinant human LTα1/β2 protein | R and D Systems Inc, Minneapolis, MN | | |
| Chemical compound, drug | CellTracker Orange CMTMR | Life Technologies/GIBCO | | |

*Continued on next page*

*Continued*

| Reagent type (species) or resource | Designation | Source or reference | Identifiers | Additional information |
|---|---|---|---|---|
| Chemical compound, drug | S1PR-antagonist FTY720 | Cayman Chemical, Ann Arbor, MI | | |
| Chemical compound, drug | S1PR1-antagonist W146 | Cayman Chemical, Ann Arbor, MI | | |
| Chemical compound, drug | S1PR3-antagonist TY52156 | Tocris Bioscience, Bristol | | |
| Other | ALZET Osmotic Pumps 1003D | DURECT Corp., Cupertino, CA | | |
| Software, algorithm | IMARIS software version 8.2 | Bitplane AG, Zurich | RRID: SCR_007370 | |

## Mice

*Lyve1*[CRE], *Spns2*[f/f] and *Rosa26LSL-tdTomato* (Ai9) mice have been previously described (*Pham et al., 2010*; *Fukuhara et al., 2012*; *Madisen et al., 2010*). *Lyve1*[CRE] mice were crossed to the *Spns2*[f/f] and the *Rosa26LSL-tdTomato* (Ai9) line in order to generate *Lyve1;Spns2*[Δ/Δ] and *Lyve1;tdTomato* mice, respectively. *Lyve1*[CRE] and *Rosa26LSL-tdTomato* (Ai9) mice were purchased from Charles River Laboratories, Inc, Wilmington, MA, USA, and C57BL/6 (wildtype) mice were bought from CLEA Japan, Inc, Tokyo. Unless otherwise stated, we used age-matched female and male mice that were between 8 and 14 weeks of age in all our experiments. All mice were housed and bred under specific pathogen-free conditions at animal facilities of the Immunology Frontier Research Center, Osaka University. All of the experimental procedures comply with 'National Regulations for the Care and Use of Laboratory Animals', and were approved by the 'Animal Care and Use Committee' of Osaka University, (Approval Nr.: 30-037-020).

## Tamoxifen-induced recombination

*Cdh5*[CRE-ERT2] (*Okabe et al., 2014*) and *S1pr1*[f/f] (*Allende et al., 2003*) mice were bred in order to develop *Cdh5*[CRE-ERT2];*S1pr1*[Δ/Δ] mice. Three weeks after birth homologous recombination in *Cdh5*[CRE-ERT2];*S1pr1*[Δ/Δ] mice was induced by i.p. injections of 50 µg/g body weight 4-hydroxytamoxifen (4-OHT, Sigma) for four consecutive days.

## Antibodies

Anti-PNAd (MECA-79) and anti-MAdCAM-1 (MECA-367) mAbs were purified from the ascites of mice inoculated *i.p.* with the hybridoma using a size-exclusion column with size-exclusion resin (Toyopearl TSK HW55; Tosoh, Japan). Purified MECA-79 was labelled with the Alexa Fluor 488 or Alexa Fluor 594 Protein Labeling Kit (Thermo Fisher Scientific, Inc, Waltham, MA, USA). Purified Anti-CD49d (PS/2) and anti-CD11a (M17/4) antibodies (used for saturation of $\alpha_4$ and $\alpha_L$ integrins in vivo) were obtained from BioXcell, West Lebanon, NH, USA. Anti–collagen IV antibody (LB-1403) was purchased from Cosmo Bio Co., Ltd, Tokyo, Japan. Fluorochrome-conjugated antibodies to mouse CD4 (RM4-5), IgA (C10-3) and CCR7 (4B12), we sourced from BD Biosciences, San Jose, CA, USA. Anti-mouse CD45 (30-F11), CD8 (53–6.7), IgM (II/41), IgD (*Adams and Alitalo, 2007*; *von Andrian, 1996*; *Mionnet et al., 2011*; *Mebius et al., 1996*; *Moussion and Girard, 2011*; *Browning et al., 2005*; *Wendland et al., 2011*; *Chyou et al., 2011*; *Kumar et al., 2015*; *Herzog et al., 2013*; *Fukuhara et al., 2012*; *Kawahara et al., 2009*; *Osborne et al., 2008*; *Hisano et al., 2012*; *Mendoza et al., 2012*; *Nagahashi et al., 2013*), CD23 (B3D4) and LYVE1 (ALY7) antibodies we obtained from eBioscience, Inc, San Jose, CA, USA. Anti-mouse CD45.1 (A20), CD3 (145–2 C11), B220 (RA3-6B2), CD21/35 (7E9), I-A/I-E (MHC-II, clone M5/114.15.2) and CD11c (N418) we purchased from BioLegend, San Diego, CA, USA. Anti-mouse fibroblast marker (ERTR7) and anti-mouse EDG-1 (H-60) we received from Santa Cruz Biotechnology, Inc, Dallas, TX, USA. Anti-mouse pAkt antibody and rabbit IgG isotype control were purchased from Cell Signaling Technology, Danvers, MA, USA.

## Flow-cytometry

Single-cell suspensions of 2x inguinal, 2x axial and 2x brachial LNs, spleen and thymus were obtained by mincing the organs gently through a 40 µm nylon mesh. Total BM cells were flushed out of 2x femurs and 2x tibiae of a single mouse with ice-cold FACS-buffer (PBS +4%FCS +5 mM EDTA). 500 µl of peripheral blood was collected from the vena cava, whereas lymph fluid was collected with a fine borosilicate glass microcapillary pipette (Sutter Instrument, Novato, CA, USA) from the *cysterna chyli*, according to a previously published protocol (*Matloubian et al., 2004*). Erythrocytes of the spleen and peripheral blood were removed by hypotonic lysis with ACK-lysis buffer. Cells were incubated with anti-mouse CD16/CD32 followed by staining with the fluorochrome-conjugated antibodies in FACS-buffer. Dead cell discrimination was performed by 7-AAD (BD Biosciences). Samples were analyzed on a FACSCanto II flow cytometer or sorted with a FACSAria II cell sorter (both BD Biosciences). Data were analyzed with FlowJo software (TreeStar, Ashland, OR, USA).

## TUNEL assay

Flow-cytometric assessment of apoptosis in $CD45^-/CD31^+/PNAd^+$ high–endothelial cells was performed using the Fluorescein In Situ Cell Death Detection Kit (Roche) according to the manufacturer's protocol upon staining with anti-CD45 and anti-PNAd antibodies.

## Lymphocyte homing assays

In order to quantify lymphocyte immigration rates into secondary lymphoid organs, $3 \times 10^7$ total splenocytes of wildtype congenic (CD45.1+) donor mice were injected *i.v.* into recipient mice upon hypotonic erythrocyte lysis. Two hours after injection, 2x inguinal, 2x axial and 2x brachial LNs were collected from recipient mice and assessed for lymphocyte immigration by FACS (*Figure 2C*). In order to quantify lymphocyte egress rates from secondary lymphoid organs $3 \times 10^7$ total splenocytes of wildtype congenic (eGFP+) donor mice were injected *i.v.* into recipient mice upon hypotonic erythrocyte lysis. After an equilibration time of 48 hr lymphocyte entry to secondary lymphoid organs was blocked by *i.v.* injection of 100 µg per mouse of neutralizing antibodies against integrin $\alpha_4$ (PS/2) and $\alpha_L$ (M17/4). Lymphoid organs were collected from recipient mice at 0 hr,and 20 hr after integrin saturation, and total cell numbers of graft lymphocytes were monitored by FACS.

## In vitro differentiation and in vivo migration of mature BMDCs

BMDCs were obtained based on a previously published protocol (*Lutz et al., 1999*). In brief, $2 \times 10^6$ total BM cells were cultivated for 10 days in 10 ml differentiation media (RPMI-1640 (Sigma, St. Louis, MO, USA), supplemented with 10 ng/ml GM-CSF (PeproTech, Rocky Hill, NJ, USA), 10% FCS (PAA laboratories GmbH, Pasching, Austria) 100 ug/ml penicillin/streptomycin, 2 mM L-glutamine, MEM non-essential amino acids, 1 mM sodium pyruvate and 0.05 mM β-mercaptoethanol (all from Life Technologies/GIBCO, Carlsbad, CA, USA) in a 10 cm petri dish. On day 3 additional 10 ml of differentiation media was added to the culture. On day 6, day 8 and day 10, 10 ml of differentiation media was substituted with fresh media and retrieved cells were returned into the culture. Complete maturation was induced on day 11 by transferring 10 ml of the original culture into a 10 cm cell culture dish. The remaining floating cells of the original culture were harvested by gentle pipetting. Upon centrifugation cells were added in 10 ml fresh differentiation media supplemented with 5 ng/ml GM-CSF + 1 µg/ml LPS (Sigma) and further differentiated for 24 hr to 36 hr. Analysis of cells obtained with this procedure revealed that >90% expressed CD11c. Mature BMDCs were labelled with CellTracker Orange CMTMR (5-[and-6]-[{(4-chloromethyl)benzoyl}amino] tetramethylrhodamine) (Life Technologies/GIBCO) [10 µm] in RPMI-1640 for 15 min at 37°C in a water bath. $1 \times 10^6$ of CMTMR-labelled BMDCs/footpad were injected into the footpad of recipient mice. Migration of mature BMDCs into popliteal LNs was confirmed by IHC of frozen sections 24 hr after footpad injection.

## Treatment with S1PR-antagonists in vivo

The non-specific S1PR-antagonist FTY720 (Cayman Chemical, Ann Arbor, MI, USA) was solved in DMSO (vehicle) and diluted in PBS containing 30% BSA prior to *i.p.* injection. The S1PR1-antagonist W146 (Cayman Chemical) and the S1PR3-antagonist TY52156 (Tocris Bioscience, Bristol, UK) were solved in deionized water containing 10 mM $Na_2CO_3$ and 2% (2-hydroxypropyl)-beta-cyclodextrin

(W146) or a 1:1 mixture of DMSO and polyethylene glycol 400 (TY52156). Mice were pre-treated with 3 mg/kg FTY20 or vehicle (control) 48 hr before subcutaneous BMDC injection into the footpad every 24 hr and received an additional FTY720 or vehicle (control) injection at BMDC-transfer (*Figure 5 (A)*). W146 and TY52156 or their respective vehicle (control) were continuously applied (13.2 µg/h) by osmotic pumps (1003D, DURECT Corp., ALZET Osmotic Pumps, Cupertino, CA, USA) implanted *i.p.* 48 hr before BMDC-injections (*Figure 5 (E)*).

## Rescue of impaired HEVs in vivo

Three weeks old mice received weekly injections of PBS (*Spns2*[f/f] and *Lyve1;Spns2*[Δ/Δ] mice) or agonistic anti-LTβr (5G11) antibody (20 µg/mouse; BioLegend) and recombinant human LTα1/β2 protein (10 µg/mouse; R and D Systems) for a total of 10 weeks. LNs of mice were analyzed by IHC for Lyve-1[+] LECs, PNAd[+] HEVs and ERTR7[+] fibroblastic tissue networks.

## Systemic depletion of DCs in vivo

XCR1-DTRvenus (*Yamazaki et al., 2013*) and CD11c-DTR (*Jung et al., 2002*) mice were injected i.p. with diphtheria toxin (DT, Sigma) or PBS as a control vehicle at a dose of 25 ng/g body weight (XCR1-DTR[venus]) or 6 ng/g body weight (CD11c-DTR) every second day for a total of eight days. The efficiency of DC depletion was controlled by staining against CD11c and FACS analyses (data not shown).

## Immunohistological analysis

Double immunostaining of fresh 6 µm-thick cryosections was performed as described previously (*Kitazawa et al., 2015*; *Sawanobori et al., 2014*). Briefly, cryosections were fixed with acetone and formol-calcium solution, then monoclonal antibodies against PNAd or other molecules were stained blue using alkaline phosphatase conjugated secondary antibodies and Vector Blue substrate (Vector Laboratories, Burlingame, CA, USA). Additionally, type IV collagen, which reveals the framework of tissues, was stained brown with a horseradish peroxidase conjugated goat anti-rabbit F(ab')two fragment (MP Biomedicals, Santa Ana, CA, USA) and 3,3'-diaminobenzidine substrate (Dojindo Molecular Technologies, Kumamoto, Japan). Photomicrographs were captured with a Microphot-FX microscope (Nikon, Tokyo, Japan) and a DP26 digital camera (Olympus, Tokyo, Japan). Anti-CD3 and anti CD11c were stained with an alkaline phosphatase labelled goat anti-hamster IgG secondary antibody (Jackson ImmunoResearch Laboratories, Inc West Grove, PA, USA) and anti-IgD, anti-IgA and anti-MAdCAM-1 were developed with an alkaline phosphatase conjugated or horseradish-peroxidase-labelled goat anti-rat IgG secondary antibody (Jackson ImmunoResearch Laboratories, Inc). Secondary immunostainings of the anti-PNAd antibody was performed with an alkaline phosphatase conjugated or horseradish-peroxidase-labelled anti-rat IgM antibody (Jackson ImmunoResearch Laboratories, Inc). The biotinylated anti-MAdCAM-1 and anti-B220 antibody was developed with alkaline phosphatase conjugated or horseradish-peroxidase-labelled streptavidin (Sigma).

## Immunofluorescent analysis

Lymphoid organs were snap frozen in O.C.T compound (Sakura Finetek Inc, Tokyo, Japan) and cut into 10-µm-thick frozen sections. The sections were blocked in 10% FCS/PBS containing mouse γ-globulins (20 µg/ml, Sigma) upon acetone, methanol (anti-EDG-1) or 4% PFA (10 min.) followed by methanol (anti-pAkt) fixation, and stained with Alexa Fluor 405, Alexa Fluor 488, Alexa Fluor 594 or Alexa Fluor 647 labelled antibodies. Fluorescence images were obtained using an Olympus Fluoview FV1000-D confocal laser scanning microscope.

## Whole-mount immunofluorescent analysis

Timed mating of mice was performed and iLNs were collected from embryos of E16.5 and E18.5. LNs were immersed for 2.5 hr in 4% PFA followed by a wash in 4% glycine/PBS and a subsequent fixation in 50%, 75% and 100% MeOH. Samples were blocked in 10% FCS/PBS containing mouse γ-globulins (20 µg/ml, Sigma) and 0.1% TritonX-100. The LNs were stained with anti-PNAd-Alexa Fluor 594, anti-MAdCAM-1-Alexa Fluor 488 and anti-LYVE1-Alexa Fluor 647 antibodies. Fluorescence images were obtained as described above.

## Image analysis

Quantification of raw image data was performed using IMARIS software version 8.2 (Bitplane AG, Zurich, Switzerland). The total number of DCs (CMTMR$^+$) or the mean fluorescence intensity of the CCL21 signal of areas in a defined radius around HEVs (PNAd$^+$) was determined using the automated spot (DCs) and surface (HEVs) detection tools by taking advantage of the distance transformation option.

## CCL21 staining

10 μm thick cryosections of pLNs were fixed in acetone and blocked in 10% FCS/PBS containing mouse γ-globulins (20 μg/ml, Sigma). Sections were first stained with goat anti-mouse CCL21 (AF457; R and D Systems Inc, Minneapolis, MN, USA) or normal goat IgG and biotinylated anti-PNAd antibody, before they were blocked with rabbit IgG and stained with horseradish-peroxidase-labelled rabbit anti-goat IgG antibody and Alexa Fluor 488 conjugated Tyramide stock solution from the Tyramide signal amplification kit (Thermo Fisher Scientific, Inc). Upon TSA signal amplification sections were additionally developed with Alexa Fluor 594-labelled streptavidin before mounting.

## High-endothelial cell culture and quantification of CCL21

pLNs (cervical, brachial, axillary and inguinal) were collected from wildtype C57BL/6, gently minced and digested with 1 mg/ml collagenase type I (Worthington Biochemical Corp., Lakewood, NJ, USA) and 2 μg/ml DNase I (Roche, Basel, Switzerland). The single-cell suspension was stained with anti-CD45 and anti-PNAd antibodies, and CD45$^-$/CD31$^+$/PNAd$^+$ high-endothelial cells were collected by FACS. $4 \times 10^4$ high-endothelial cells were cultured in 100 μl serum-free EC-growth media (HuMedia-EG2; Kurabo, Osaka, Japan) with or without 10 μM FTY720 (non-specific S1PR antagonis) or W146 (S1PR1-specific antagonist) for 4 hr without S1P. After four hours, the cell culture supernatant was collected and high-endothelial cells were cultured further with media supplemented with charcoal-stripped FCS (homemade and checked by LC-MS/MS for depletion of S1P). One half of the cultures were cultivated without S1P, whereas the other half of the cultures was cultivated in the presence of 1 μM D-erythro-sphingosine-1-phosphate (Avanti Polar Lipids, Inc, Alabaster, AL, USA) for additional 20 hr. The concentration of CCL21 in the cell culture supernatants collected after four or 20 hr was quantified by ELISA using the mouse CCL21 DuoSet ELISA kit (R and D Systems, Minneapolis, MN, USA) according to the manufacturer's protocol.

## Quantification of S1P

The concentration of S1P was quantified by LC-MS/MS as described previously with minor modifications (*Saigusa et al., 2012*). Blood plasma or lymph samples (10 μL) were mixed with 8-times volume of methanol (80 μL) and equal volume of an internal standard, 1 μM dihydro-S1P (10 μL), in a 1.5 mL siliconized plastic microtube (Watson, USA). The tube was treated with an ultrasonic bath for 10 min and centrifuged at 21,500 g for 10 min at 4°C. The deproteinized supernatant was filtered and 10 μL of the filtrate was subjected to the LC-MS/MS. The LC-MS/MS system was equipped with a NANO-SPACE SI-II HPLC (Shiseido, Japan) and a TSQ Quantum Ultra triple quadrupole mass spectrometer (Thermo Fisher Scientific, USA) with a heated-electrospray ionization-II (HESI-II) source. LC separation was performed using a Capcell Pak C18 ACR (1.5 mm i.d. ×100 mm, 3 μm, Shiseido) with a gradient elution of the two mobile phase (5 mM ammonium formate (HCOONH4)–H2O (pH 4.0) (A) and 5 mM HCOONH4–H2O/acetonitrile (CH3CN) (a volume ratio of 5/95; apparent pH 4.0) (B) at a flow rate of 150 μL/min. The gradient was initial elution with 50% B, followed by linear gradient to 95% B (from 0.2 to 2.8 min), 95% B isocratic (from 2.8 to 5.0 min), return to the initial 50% B condition (from 5.0 to 5.1 min) and then equilibrium (from 5.1 to 6.5 min) until next sample injection. Elute was continuously ionised and positive ions were detected by the mass spectrometer with a selected reaction monitoring mode using the following transitions: m/z 368.2 → 270.2 for dihydro-S1P and m/z 380.3 → 264.2 with a collision energy of 18 V for S1P. Ions were fragmented using a collision Argon pressure of 1.5 mTorr with collision energy of 16 V (dihydro-S1P) and 18 V (S1P). Mass chromatogram was obtained and analyzed by a Xcalibur software (Thermo Fisher Scientific) and peak area ratio between analyte and internal standard was used for quantification by using a standard curve.

## Transmission electron microscopy

Mice were perfused with a solution of 2% glutaraldehyde + 4% paraformaldehyde in 0.1 M phosphate buffer (pH 7.4) through the left ventricle. Removed LNs were postfixed by 2% osmium tetroxide and dehydrated in a graded ethanol series, and then embedded in Quetol 812 epoxy resin. Ultrathin sections were stained with 2% uranyl acetate and Reynold's lead citrate, and examined using a JEM-1230 electron microscope (JEOL Ltd., Tokyo, Japan).

## RNA isolation and quantitative RT-PCR

Total RNA was isolated using the RNeasy mini kit (Qiagen, Hilden, Germany) or the NucleoSpin RNA Blood kit (Macherey-Nagel GmbH and Co. KG, Düren, Germany), and cDNA was synthesized with oligo(dt) primers and Superscript-III-RT (Invitrogen, Carlsbad, CA, USA). Expression of mRNAs was quantitatively assessed by real-time RTF-PCR using the SYBR Premix Ex Taq (Tli RNaseH Plus; Takara bio, Inc, Kusatsu. Japan) in a Thermal Cycler Dice Real Time System II (Takara bio, Inc). Each sample was assayed in duplicates for every run and results were normalised against Hprt or Gapdh mRNA expression.

The PCR primer pairs used here were:

| Gene | Forward (5' – 3') | Reverse (5' – 3') |
|------|-------------------|-------------------|
| Spns2 | CCATCCTGAGTTTAGGCAACG | GATCACCTTTCTATTGAAGCGGT |
| CCL19 | CTGCCTCAGATTATCTGCCAT | AGGTAGCGGAAGGCTTTCAC |
| CCL21 | AAGGCAGTGATGGAGGGG | CGGGTAAGAACAGGATTG |
| CXCL13 | CATAGATCGGATTCAAGTTACG | TCTTGGTCCAGATCACAACTTCA |
| GlyCAM-1 | AGAATCAAGAGGCCCAGGAT | TGGGTCTTGTGGTCTCTTCCA |
| CD34 | AAACGGCCATTCAGCAAGACAACA | TCGCCACCCAACCAAATCACAAG |
| MadCAM-1 | GTCCTGCACGGCCCACAACAT | CCAGTAGCAGGGCAAAGGAGAG |
| GlcNAc6ST-2 | GGCAAGCAGAAGGGTTAGG | CTGGGAACCCAGGAACATC |
| FucT-VII | CAGATGCACCCTCTAGTACTCTGG | TGCACTGTCCTTCCACAACC |
| ENPP2 | TTGTCCGCCCTCCGTTAATC | ACAGGACCGCAGTTTCTCAA |
| CD31 | TCCCTGGGAGGTCGTCCAT | GAACAAGGCAGCGGGGTTTA |
| VCAM-1 | GGATCGCTCAAATCGGGTGA | GGTGACTCGCAGCCCGTA |
| ICAM-1 | GGGAATGTCACCAGGAATGT | GCACCAGAATGATTATAGTCCA |
| VE-cadherin | TCCTCTGCATCCTCACTATCACA | GTAAGTGACCAACTGCTCGTGAAT |
| LTbR | TGGTGCTCATCCCTACCTTCA | TTCTCTCTATCCTCTCCCCCAG |
| Gapdh | CCTCGTCCCGTAGACAAAATG | TCTCCACTTTGCACTGCAA |
| Hprt | AGGTTGCAAGCTTGCTGGT | TGAAGTACTCATTATAGTCAAGGGCA |

## Statistical analysis

All the statistical analysis on the data was performed using GraphPad Prism software version 5.03. Statistical significance of differences between two groups with normally distributed data was determined using a Student's unpaired two-tailed $t$-test.

Sample sizes were chosen according to prior experiences with specific mouse lines.

More detailed descriptions of the experimental procedures can be found in the supplementary information.

## Acknowledgements

We thank Dr. Junichi Kikuta, Norie Yoshizumi, Mai Shirazaki, Masami Kasaoka, Yumiko Mayjima, Dr. Hiroaki Hemmi, Furi Fukuda, Dr. Tetsuya Honda, and Dr. Tomoko Hirano for technical assistance. We thank Dr. Kuniyuki Kano for providing charcoal-stripped FCS. We thank Dr. Yoshiaki Kubota for kindly providing Cdh5[CRE-ERT2] mice, and we are grateful to Dr. Richard L Proia and Dr. Sylvio Gutkind for kindly giving us S1pr1[f/f] mice. Furthermore, we thank Dr. Ronald Germain, Dr. Takaharu

Okada, Dr. Alison Hobro, Dr. Sidonia Fagarasan, Dr. Fritz Melchers, Dr. Tsuneyasu Kaisho, and Dr. Kenji Kabashima for active discussions, technical advice and critical suggestions for the manuscript. This work was supported by Grants-in-Aid for Scientific Research (A) from the Japan Society for the Promotion of Science (JSPS; JP25253070 and JP16H02619 to MI); Grants-in-Aid for Young Scientists (B) from the JSPS (JP26860329 to SS); grants from the Takeda Science Foundation (to MI); from the Uehara Memorial Foundation (to MI); from the Kanae Foundation for the Promotion of Medical Sciences (to MI); and from the Kishimoto Foundation (to SS).

## Additional information

### Funding

| Funder | Grant reference number | Author |
|---|---|---|
| Japan Society for the Promotion of Science | Grants-in-Aid for Scientific Research (A) JP25253070 | Masaru Ishii |
| Takeda Science Foundation | Research Grant | Masaru Ishii |
| Kanae Foundation for the Promotion of Medical Science | Research Grant | Masaru Ishii |
| Uehara Memorial Foundation | Research Grant | Masaru Ishii |
| Japan Society for the Promotion of Science | Grants-in-Aid for Young Scientists (B) JP26860329 | Szandor Simmons |
| Kishimoto Foundation | Postdoctoral Fellowship | Szandor Simmons |
| Japan Society for the Promotion of Science | Grants-in-Aid for Scientific Research (A) JP16H02619 | Masaru Ishii |

The funders had no role in study design, data collection and interpretation, or the decision to submit the work for publication.

### Author contributions

Szandor Simmons, Conceptualization, Resources, Data curation, Formal analysis, Supervision, Validation, Investigation, Visualization, Methodology, Writing—original draft, Project administration, Writing—review and editing; Naoko Sasaki, Eiji Umemoto, Yutaka Uchida, Shigetomo Fukuhara, Yusuke Kitazawa, Michiyo Okudaira, Asuka Inoue, Kazuo Tohya, Keita Aoi, Investigation, Methodology; Junken Aoki, Naoki Mochizuki, Kenjiro Matsuno, Kiyoshi Takeda, Supervision, Methodology; Masayuki Miyasaka, Conceptualization, Supervision, Methodology, Writing—original draft; Masaru Ishii, Conceptualization, Supervision, Funding acquisition, Methodology, Writing—original draft, Project administration, Writing—review and editing

### Author ORCIDs

Szandor Simmons https://orcid.org/0000-0001-9619-0942
Naoki Mochizuki http://orcid.org/0000-0002-3938-9602

### Ethics

Animal experimentation: All of the experimental procedures comply with "National Regulations for the Care and Use of Laboratory Animals", and were approved by the "Animal Care and Use Committee" of Osaka University (Approval Nr.: 30-037-020).

### Decision letter and Author response

Decision letter https://doi.org/10.7554/eLife.41239.017
Author response https://doi.org/10.7554/eLife.41239.018

## Additional files

### Supplementary files
• Transparent reporting form
DOI: https://doi.org/10.7554/eLife.41239.015

### Data availability
All data generated or analysed during this study are included in the manuscript and supporting files.

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
