## [Decision Letter]

Thank you for submitting your article "High-endothelial cell-derived S1P regulates dendritic cell localization and vascular integrity in the lymph node" for consideration by *eLife*. Your article has been reviewed by three peer reviewers, including Michael L Dustin as the Reviewing Editor and Reviewer #1, and the evaluation has been overseen by Tadatsugu Taniguchi as the Senior Editor. The following individual involved in review of your submission has agreed to reveal their identity: Shannon Turley (Reviewer #3).

The reviewers have discussed the reviews with one another and the Reviewing Editor has drafted this decision to help you prepare a revised submission.

The authors contend that autocrine S1PR1 signaling in HEV is required for optimal CCL21 expression leading to positioning of endogenous and transferred DC within 40 μm of HEV. This is an important concept as positioning antigen positive DC near HEV could increase the rate of newly entering T cells encountering antigens on DC. A limitation of the methodology is that both the Cre drivers used in the study also will delete Spns2 or S1PR1 on blood and lymphatic endothelial cells. Thus, the authors are put in the position of needing to exclude lymphatic endothelial cells playing a role in the phenotype.

Essential revisions:

1) The authors make the statement that" In the conditional VEcad-CRE-ERT2- S1PR1^Δ/Δ^ mice we deleted S1PR1 on all blood vascular cells, including HEVs, postnatally by tamoxifen administration." without offering proof of this. Since the performance of Cre-ERT2 driven by tissue specific promoters will vary for any given floxed locus, it important to demonstrate the efficacy of the knockout at the protein level in single cells (flow cytometry). These data, which they must have based on the statement, should be added to supplement the paper. It would be ideal to examine both blood and lymphatic endothelial cells and potentially to look at S1PR1, CCL21, Lyve1 and PNAd if possible.

2) In the VEcad-Cre-ERT2- S1PR1^fl/fl^, is there also a defect in lymphocyte numbers in blood, lymph node or spleen? You would not expect the T cell survival defect in these mice, but is there evidence of a lymphocyte entry defect, or only the DC positioning. These mice should be characterized better. If this is already described somewhere, please discuss these findings in relation the Lyve1-cre Spns2 CKO. If the defects are restricted to the DC localization this would make a strong case that S1PR1 on HEV is mainly important for DC localization and not so much for T cell entry function. In the Spns2 deficient mice the appearance of defects in HEV may be more related to lack of T cell recirculation.

3) It is very unfortunate that the HEV specific cre mice are not available. However, short of remaking these, can the authors offer a similar quantitative analysis of DC positions in relations to cortical lymphatics in the Lyve-1-cre Spns2^fl/fl^ and VE-card-Cre-ERT2-S1PR1^fl/fl^ mice to show that the change in DC location is specific to HEV and that there is no impact in relation to cortical lymphatics. The HEV and cortical lymphatics are sufficiently distinct to make this analysis meaningful.

---

## [Author Response]

Essential revisions:1) The authors make the statement that" In the conditional VEcad-CRE-ERT2- S1PR1^Δ/Δ^ mice we deleted S1PR1 on all blood vascular cells, including HEVs, postnatally by tamoxifen administration." without offering proof of this. Since the performance of Cre-ERT2 driven by tissue specific promoters will vary for any given floxed locus, it important to demonstrate the efficacy of the knockout at the protein level in single cells (flow cytometry). These data, which they must have based on the statement, should be added to supplement the paper. It would be ideal to examine both blood and lymphatic endothelial cells and potentially to look at S1PR1, CCL21, Lyve1 and PNAd if possible.

We thank the reviewer for raising this valid point. We have now incorporated these findings in the new Figure 6—figure supplement 1A-B and as follows:

“The results of the S1PR-antagonist experiments and the S1PR1 expression analyses guided us to conditionally delete *S1pr1* on high-endothelial cells and to analyse if DCs are able to co-localize with *S1pr1*-deficient high-endothelial cells in pLNs. Therefore, we opted to generate *Cdh5*^CRE-ERT2^;*S1pr1*^Δ/Δ^ mice. Flow-cytometry revealed the efficient deletion of S1pr1 in HEVs and LECs in the conditional *Cdh5*^CRE-ERT2^;*S1pr1*^Δ/Δ^ mice by postnatal tamoxifen administration (Figure 6—figure supplement 1A-B).”

Due to a limited availability of Cdh5^CRE-ERT2^;S1pr1^Δ/Δ^ mice we had to prioritize the analyses of S1PR1, Lyve1 and PNAd over CCL21 and, therefore, apologize for not incorporating CCL21 expression levels in our analyses.

2) In the VEcad-Cre-ERT2- S1PR1^fl/fl,^ is there also a defect in lymphocyte numbers in blood, lymph node or spleen? You would not expect the T cell survival defect in these mice, but is there evidence of a lymphocyte entry defect, or only the DC positioning. These mice should be characterized better. If this is already described somewhere, please discuss these findings in relation the Lyve1-cre Spns2 CKO. If the defects are restricted to the DC localization this would make a strong case that S1PR1 on HEV is mainly important for DC localization and not so much for T cell entry function. In the Spns2 deficient mice the appearance of defects in HEV may be more related to lack of T cell recirculation.

We thank the reviewers for the helpful suggestion to incorporate quantifications of lymphocyte numbers in peripheral lymphoid tissues into the manuscript. We added to the revised version of our manuscript the new Figure 6—figure supplement 1C-D and the following sentence:

“*…Cdh5*^CRE-ERT2^;*S1pr1*^Δ/Δ^ mice did not show any differences in CD4 or CD8 single-positive T-cell and mature rec. B-cell numbers in peripheral blood or pLNs (Figure 6—figure supplement 1C-D).”

These observations are in line with previous descriptions of lymphocyte numbers in Cdh5^CRE-ERT2^;S1pr1^Δ/Δ^ mice published by Timothy Hla’s research group. Blaho et al., 2015, showed that “…lymphocytosis was not caused by a loss of endothelial cell S1P1 signaling, since inducible endothelial cell-specific deletion of S1pr1 (S1P1 ECKO) did not affect blood lymphocyte numbers” in S1pr1^f/f^ Cdh5-Cre-ER^T2^ (ECKO) mice. Furthermore, Galvani et al., 2015, described that “…monocyte (CD11b^+^, CD115^+^, Ly6G^−^), neutrophil (CD11b^+^, Ly6C^int^, Ly6G^hi^), T lymphocyte (CD4^+^ and CD8^+^), and B lymphocyte (CD19^+^) numbers were similar in Apoe^−/−^ S1pr1ECKO and Apoe^−/−^ S1pr1 wild-type mice.” Therefore, we conclude that autocrine activation of S1PR1-signalling on HEV is mainly important for DC-localization and to a much lesser extent for the regulation of T-cell immigration from the blood circulation into pLNs. A respective paragraph has been added to the Discussion section (sixth paragraph).

3) It is very unfortunate that the HEV specific cre mice are not available. However, short of remaking these, can the authors offer a similar quantitative analysis of DC positions in relations to cortical lymphatics in the Lyve-1-cre Spns2^fl/fl^ and VE-card-Cre-ERT2-S1PR1^fl/fl^ mice to show that the change in DC location is specific to HEV and that there is no impact in relation to cortical lymphatics. The HEV and cortical lymphatics are sufficiently distinct to make this analysis meaningful.

We thank the reviewers for this thoughtful comment, which helped us to underline our point that DC positioning is HEV- and not LEC-dependent. We have now incorporated the DC-positioning analyses relative to cortical lymphatics in the new Figure 6—figure supplement 1E-G and into the text as follows:

“Indeed, when we performed WT BMDC injections into the footpad of *Cdh5*^CRE-ERT2^;*S1pr1*^Δ/Δ^ mice (Figure 6C) and analysed HEV-DC interactions 24 hours later, we could detect a remarkable C impairment of total DC-numbers within a restricted area of 0 – 60 μm around the basal lamina of HEVs (Figure 6D-F), while no differences in DC-positioning close to cortical lymphatics were observed (Figure 6—figure supplement 1E-G).”